# Unveiling the cell-type-specific landscape of cellular senescence through single-cell transcriptomics using *SenePy*

Mark A. Sanborn [1,2] ✉, Xinge Wang [1,2,3], Shang Gao[1,2,3], Yang Dai [2,3] & Jalees Rehman [1,2,3,4] ✉

Senescent cells accumulate in most tissues with organismal aging, exposure to stressors, or disease progression. It is challenging to identify senescent cells because cellular senescence signatures and phenotypes vary widely across distinct cell types and tissues. Here we developed an analytical algorithm that defines cell-type-specific and universal signatures of cellular senescence across a wide range of cell types and tissues. We utilize 72 mouse and 64 human weighted single-cell transcriptomic signatures of cellular senescence to create the *SenePy* scoring platform. *SenePy* signatures better recapitulate in vivo cellular senescence than signatures derived from in vitro senescence studies. We use *SenePy* to map the kinetics of senescent cell accumulation in healthy aging as well as multiple disease contexts, including tumorigenesis, inflammation, and myocardial infarction. *SenePy* characterizes cell-type-specific in vivo cellular senescence and could lead to the identification of genes that serve as mediators of cellular senescence and disease progression.

Aging is a key risk factor for many chronic diseases[1]. One biological manifestation of organismal aging is cellular senescence (CS), a phenomenon characterized by permanent cell cycle arrest, impaired homeostatic cellular function, and activation of the senescence-associated secretory phenotype (SASP), which involves the release of pro-inflammatory proteins, proteases, and other bioactive paracrine factors[2]. Senescent cells accumulate in tissues with increasing organismal age, but senescent cells are found even in young organisms and can accrue prematurely due to exogenous stressors[3,4]. Accumulated senescent cells contribute to sterile inflammation, tissue remodeling, and local dysfunction, which ultimately drives various pathologies[5]. CS contributes to a wide array of chronic diseases, including cardiovascular disease, neurodegeneration, and diabetes[6–8]. The senescent cell burden in aged organisms also contributes to unchecked inflammation and poor outcomes in acute diseases, such as coronavirus infection[9]. Targeted clearance of senescent cells with senolytics can mitigate disease severity and increase healthspan[7–10], but their elimination may

exacerbate disease in some contexts[11]. Despite the growing understanding of the role of CS in aging and various diseases, in vivo CS remains poorly phenotypically and mechanistically characterized[2,12]. Most CS markers have been identified in cultured cells subjected to experimental conditions that may not accurately represent a living system. More comprehensive markers are required to robustly study CS in living systems.

One of the biggest challenges in studying in vivo CS is the high degree of heterogeneity, as CS involves a multitude of changes in cellular function[2]. CS has been observed in numerous cell types across all major organs. Senescent cells partially lose their pre-senescence cell identities and phenotypes, suggesting that the mechanistic paths to the senescent states vary between cell types[13,14]. Cells are also subject to various cell-intrinsic and extrinsic stressors that drive CS. Telomere attrition is a well-known CS trigger, but telomere-independent DNA damage, oxidative stress, and oncogenic signaling can also induce cellular senescence[5]. Paracrine factors released in the SASP state and

[1]Department of Biochemistry and Molecular Genetics, University of Illinois, College of Medicine, Chicago, Illinois, USA. [2]Center for Bioinformatics and Quantitative Biology, University of Illinois Chicago, Chicago, Illinois, USA. [3]Department of Biomedical Engineering, University of Illinois Chicago, College of Engineering and College of Medicine, Chicago, Illinois, USA. [4]University of Illinois Cancer Center, Chicago, Illinois, USA. ✉e-mail: msanbo2@uic.edu; jalees@uic.edu

cell-surface signaling from senescent cells can induce secondary CS among otherwise healthy cells in close proximity[15,16]. Drivers of CS shift the transcriptional landscape of senescent cells[13,14], but this has been primarily studied in cultured cells. The tissue context or cell identity-specific transcriptional landscapes of CS have not been fully defined. For example, it is unclear whether the CS transcriptional programs in skin fibroblasts exposed to UV light differ from those of fibroblasts in internal organs protected from light. The heterogeneity arising from different stressors, tissues, and cell types makes it difficult to broadly apply transcriptional signatures of CS derived from in vitro cultured cells that have been removed from their in situ tissue environment. There are no universal signatures or markers of CS[2]. Even the cell cycle arrest inducer p16[ink4a] (encoded by the gene *CDKN2A*), which is widely accepted as one of the most specific markers of CS, is not always required for CS induction and its use as a sole marker in transcriptomics data is confounded by the fact that the corresponding *CDKN2A* locus encodes for multiple genes with overlapping sequence identity[17–19]. Other CS markers may be constitutively expressed in some cell types or upregulated in general with organismal age or inflammation. Recent work has utilized literature screening and transcriptomics to find a gene set that is broadly differentially abundant in senescent cells[20], but this does not account for tissue-, cell-, or stress-specific heterogeneity and may not capture all programs of CS. There remains a need to identify and characterize tissue- and cell-specific CS programs[21].

In this study, we aggregate and interrogate large-scale single-cell RNA-sequencing datasets across tissues and ages in mice and humans to define in vivo CS heterogeneity. We developed an algorithmic approach to identify cell-type-specific CS signatures. We used a p16[ink4a] reporter mouse model dataset and other transcriptomics datasets to validate our approach. We have generated the open-source Python package *SenePy* (https://github.com/jaleesr/senepy). *SenePy* allowed us to map the kinetics of CS in many tissues and cell types with respect to organismal age and in the context of disease. Using *SenePy* we were able to identify senescent cells across several tissues and examine similarities as well as tissue-specific and cell-type-specific signatures of cellular senescence.

## Results

### Known cellular senescence markers are cell-type-specific and poorly characterize in vivo cellular senescence

We examined the expression of established CS markers in comprehensive mouse and human single-cell atlases to determine their dynamics with age and their cell-type-specificity. For mouse data, we utilized the *Tabula Muris Senis*[22] resource, which is comprised of 328 K cells from 19 tissues collected between 1 and 30 months of age (Supplementary Fig. 1A). The human data utilized in this study are derived from 7 studies[23–29] that span 37 tissues from individuals aged 1 to 92 years old, altogether comprising 1.6 M cells (Supplementary Fig. 1B). We took the union of *SenMayo*[20] ($n = 125$), which is a recently published list of cellular senescence-associated genes, and a self-curated CS gene set ($n = 108$ human, $n = 110$ mouse) to establish a panel of 181 (181 human, 184 mouse) experimentally validated CS marker genes. The independently generated gene set had significant overlap with SenMayo but contained many unique genes (intersection = 52,51, hypergeometric *p*-value = $5.5 \times 10^{-97}$, $2.85 \times 10^{-79}$). This panel of known or validated CS markers served as a starting point for our downstream analyses.

To assess how this set of CS markers overlapped with cell-type-specific and universal organismal aging genes in mice, we compared them to a study that identified 76 cell-type-specific signatures and a universal aging signature ($n = 330$ genes) from *Tabula Muris Senis*[30] (Fig. 1A). Only the CS markers *Cd9, Ctnnb1, and Jun* were present in the universal organismal aging signature (defined by genes upregulated with age in half of the cell types), which can be explained by random

chance (hypergeometric *p*-value = 0.58). *Cdkn2a* (p16[ink4a] encoding gene), widely accepted as one of the most universal and specific markers of CS, was not present in any of the cell-specific organismal aging signatures. Furthermore, only 15 of the 76 cell-specific aging signatures were enriched for genes from our CS marker panel (Supplementary Fig. 2A, B). This observed cell-type-specific enrichment is not explainable by known CS dynamics and was negatively correlated to population-specific proliferation, as determined by the proportion of *Mki67* + cells (Pearson's $R = -0.24$, *p*-value = 0.04) and not correlated to cell cycle score ($R = -0.07$, *p*-value = 0.5) (Supplementary Fig. 2C). These findings suggest that a universal CS signature may be obfuscated by cellular heterogeneity and that differential expression between subpopulations of cells is not suitable for extracting CS-specific markers. Instead, we show that the proportions of cells positive for *Cdkn2a* and other CS markers increase significantly with age in *Tabula Muris Senis* (Fig. 1B), thus indicating that difference in the proportions of cells expressing specific CS-associated genes is a more useful metric for identifying dynamic CS genes than using differential expression analyses.

We next analyzed the aging dynamics of CS markers across all tissues and cell types in the mouse and human datasets with respect to the proportion of positive cells. Some of the most widely used markers of CS, such as *Cdkn2a* and *Cxcl13*, showed an overall increase in the proportion of cells expressing these genes with age (FDR *p*-values = 0.01, 0.03) but had tropism for specific tissue and cell types (Fig. 1C and Supplementary Fig. 3A). Human skin from the face, for example, had appreciable levels of *CDKN2A* + cells in young individuals and a large increase with age (Student's *t* test, one-tailed, *p*-value = 0.002) (Supplementary Fig. 3B). However, other important markers of CS, such as *CDKN1A* (p21[cip1] encoding gene), were more constitutively expressed in young and old mice and humans (Fig. 1D and Supplementary Fig. 3C).

We examined the dynamics of cellular senescence markers in 60 mouse and 50 human cell types (Supplementary Data 1, 2). Overall, the landscape of all CS markers was highly heterogeneous when stratified by tissue and cell type in both species (Fig. 1E, F). Only 58 of 1540 pairwise combinations of cell types showed significant CS marker gene overlap (Hypergeometric, FDR *p*-value < 0.05), thus highlighting how CS program transcriptional profiles differ widely between tissues and cell types. Cells from the same tissue were most likely to have significant marker overlap (Chi-square, *p*-value = $3.2 \times 10^{-12}$). While fibroblasts from different tissues showed some overlap in senescent cell marker profiles, this was not statistically enriched (Chi-square, *p*-value = 0.8). The marker found to increase dynamically with age in the largest number of human cell types was *CCL4*, but this marker was only dynamic in 18 of 50 (36%) of the cell types tested. *Ccl5* and *Ccl8* were the most universally dynamic in mice but only in 33% of cell types. *CDKN2A* was one of the most enriched CS markers in both humans and mice, but it was only dynamic in 26% of human and 32% of mouse cell types. These data indicate that there is no universal CS marker gene set for all tissues and cell types and that each cell type within each tissue takes distinct transcriptional paths to the CS state. Instead, our data maps the suitability of known CS markers in different organisms, tissues, and cells.

### De novo cell-type-specific signatures derived algorithmically from single-cell transcriptomes

We developed an unbiased computational method to identify putative CS signatures in distinct mouse and human cell populations (Fig. 2A and Supplementary Data 3, 4). These signatures were derived from 43 mouse and 45 human cell types that spanned wide organismal ages. We leveraged the prior biological knowledge that senescent cells accumulate with increasing organismal age but remain a minority population[31] to identify dynamic genes within each cell type. Using binarized positivity matrices with the selected genes and cells, we

constructed networks with hubs of genes likely to be expressed in the same cells. Genes not associated with hubs were removed as they likely represented noise or stochastic processes, which increase with organismal age but are not directly related to cellular senescence. Some cell types comprised multiple distinct signatures separated into subnetworks during network analysis (Fig. 2B). In total, we derived 72

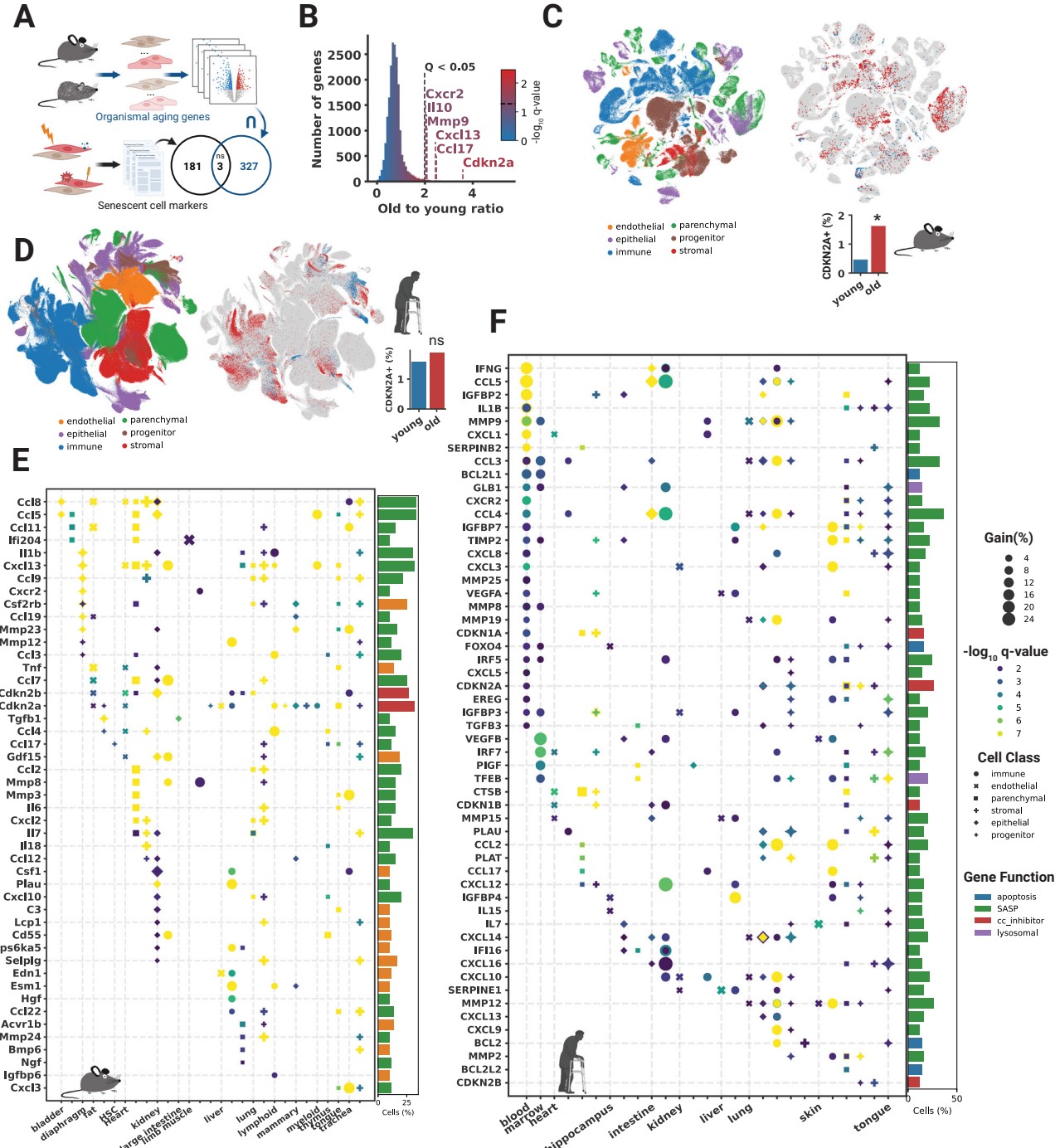

**Fig. 1 | Known markers are cell-type-specific and poorly characterize in vivo cellular senescence. A** There is an insignificant overlap between a universal organismal aging signature and previously reported senescence markers ($p = 0.58$, Hypergeometric). The universal signature is comprised of genes present in at least 50% of the cell-specific gene sets elucidated by differential expression between young and old cells previously. Created in BioRender. Rehman, J. (2025) https://BioRender.com/w72f903 **B** A histogram depicting the 24-month to 3-month ratio (old to young ratio) of all cells expressing every gene in the *Tabula Muris Senis* dataset. Genes to the right of the dashed line have a statistically significant increase based on random permutations. Statistically significant known senescence markers are labeled. **C, D** UMAPs (right) representing *CDKN2A* + (p16ink4a encoding gene) cells from the mouse and human datasets. Cells from 24-, and 30-month mice are denoted old while 1-, and 3-month mice are young. UMAPs (left) show all cells in the datasets and are labeled by broad cell classifications. Bar graphs show the percentage of *CDKN2A* + cells relative to all cells in the respective datasets. (* = FDR *p*-value = 0.011, $n = 1000$ random permutations). **E, F** Cell-specific maps of marker dynamics in mice and humans. Vertical dashed lines represent the start of a tissue and cell types from that tissue are classified and depicted by marker shape. Multiple cell types belonging to the same class are overplotted. The gain represents the percent increase of cells expressing the marker relative to young organisms. Only statistically significant genes are shown. Bar plots colored by senescence-associated gene function depict the percentage of cell populations in which the respective gene is a suitable marker.

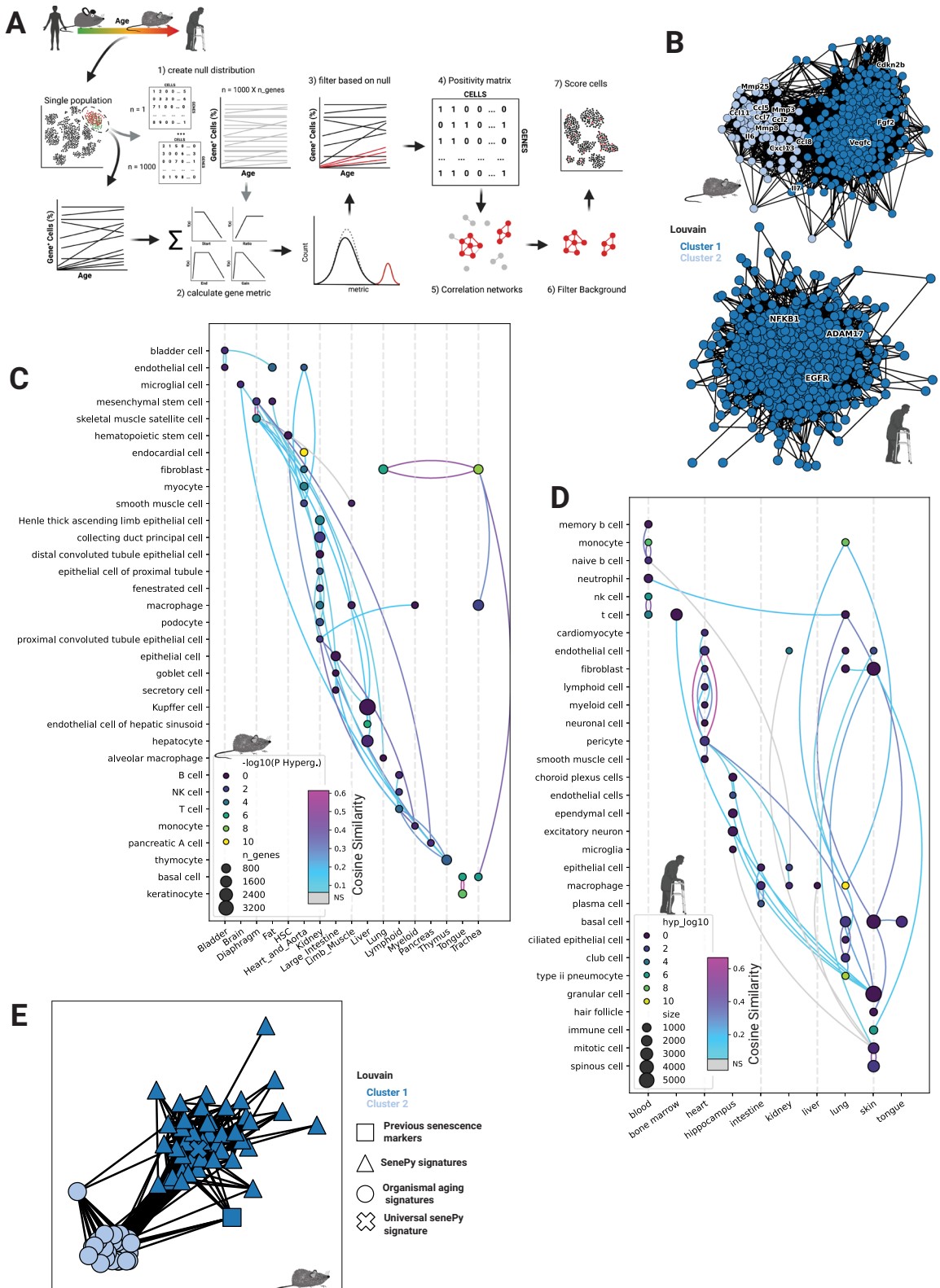

mouse and 64 human putative CS cell-type signatures (Supplementary Data 4, 5). The signatures contained both genes as well as values which indicate the degree of correlation they have to other genes within the same signature.

Cell-type signatures were highly heterogeneous, but known markers of CS were enriched in selected cell types, although not in a consistent manner (Fig. 2C, D). The signatures were enriched for known CS markers at a higher rate than gene sets derived from

differential expression analyses (Supplementary Fig. 2d). Importantly, similar gene expression signatures (Hypergeometric FDR < 0.05) were more likely to be found between cells from the same tissue (Chi-square, $p$-value = $1.7 \times 10^{-8}$) than between cells of the same cell type (Chi-square, p-value = 0.004) within *Tabula muris senis*. Signatures from cells within the same tissue had a higher average cosine similarity of 0.09 compared to 0.04 for those derived from different tissues (Mann Whitney, $p = 9.6 \times 10^{-7}$). However, there were some exceptions

**Fig. 2 | De novo cell-type-specific signatures derived algorithmically from single-cell transcriptomes. A** Overview of the algorithm used to define cell-specific signatures from mice and humans (see methods). Created in BioRender. Rehman, J. (2025) https://BioRender.com/f97q817. **B** Example signatures derived from mouse cardiomyocytes (top) and human hippocampal choroid plexus cells (bottom). Each node represents a gene and the connections represent co-positivity in cells. Connections are weighted by Pearson's R. The colors represent distinct hub signatures within the overall cell signatures. **C, D** Representative diagram of all derived signatures from mice and humans. Each dot represents a signature and is sized by its number of genes. The dot color is the respective enrichment for each signature compared to previously known senescence markers ($-\log_{10}$ Hypergeometric $P$-value). Each signature is connected to its most similar signature and the color of the connection is based on the cosine similarity. Signatures without significant overlap are connected with gray lines (Hypergeometric FDR $P$-value). **E** Network similarity analysis of mouse cell-specific novel senescence signatures and organismal aging signatures. Each shape represents a signature and lines represent significant similarity between them. Similarity (strength of connections) is defined as $-\log_{10}$(BH-corrected Hypergeometric P). The network is clustered and colored by Louvain's algorithm.

in which signatures from cell types found in multiple tissues shared high similarity. For example, senescent fibroblasts found in mouse lungs were most similar to senescent fibroblasts from mouse tracheas and share the overall highest similarity between any two mouse senescent cell type signatures. Conversely, fibroblast signatures taken from all of the tissues were not more likely to be similar to each other (Chi-square, $p$-value = 0.4). Many cell-type signatures contained overlapping genes despite the high overall signature heterogeneity (Fig. 2E). There were 368 of 903 (40%) mouse cell-type-signature pairs that shared significant hypergeometric overlap compared to 58 out of 1540 (4%) when using established markers. Based on their genetic profile, cell-type signatures clustered distinctly with each other but not with organismal aging signatures.

These observations indicate that the signatures we derived share some underlying genetic characteristics despite being highly distinct and that these signatures likely represent bona fide in vivo CS programs. Therefore, we developed the open-source *SenePy* Python software package to score single cells based on their expression of these CS signature genes. *SenePy* rapidly processes thousands of cells and provides relative senescent scores for every cell within a given population based on the selected signatures (Methods: Scoring cells using *SenePy*).

## Distinct modes and phenotypes of cellular senescence exist within the same cell types

Cell populations are exposed to multiple stressors and signals; therefore, they may harbor multiple independent modes of CS. We observed that multiple cell-type signatures consisted of correlated gene hubs that clustered during network analysis (Figs. 2B, 3B and Supplementary Fig. 4a). Of the 43 mouse cell types with derived cell-type signatures, 25 contained multiple gene subnetwork clusters separated using Louvain's algorithm, resulting in 72 total signatures (Supplementary Fig. 4a). Of these 72 gene subnetworks, 36 were statistically enriched for known CS markers (Supplementary Fig. 4b) These subnetwork clusters are hereby referred to as hubs. From 45 human cell types, 15 signatures were comprised of multiple hubs. Multiple gene signature hubs likely represent modes of CS with distinct kinetics and gene expression patterns within the same cell type. Generally, the number of senescent outlier cells was higher in aged cells (Wilcoxon signed-rank $p$-value = $2.6 \times 10^{-14}$) (Fig. 3A), but several cell types had hubs with divergent kinetics (Supplementary Fig. 4c). Aged mouse tongue keratinocytes consisted of two hubs with divergent kinetics, both enriched in established CS marker genes (hypergeometric $p$-values = $1.5 \times 10^{-7}$, $3.5 \times 10^{-2}$) (Fig. 3B), and the respective sample data contained adequate numbers of cells at each age, making it a suitable representative sample for similar signatures. Gene enrichment analysis indicated different phenotypes and functional roles between cells expressing these separate hubs (Fig. 3C, D). One of the tongue keratinocyte hubs was more proinflammatory and primarily enriched for immune cell chemotaxis, cytokine, and TNF signaling pathways, typical of the senescence-associated secretory phenotype. We have termed this "type-A" senescence. The other, "type-B", was enriched for innate immune processes and pathways that typically respond to pathogenic stimuli, for example, the NOD-like

receptor signaling pathway (Fisher's Exact FDR $p$-value = $3.5 \times 10^{-11}$). These observations suggest that these different modes of CS drive different inflammatory pathways and may contribute differentially to sterile inflammation. Both type-A/B tongue keratinocyte gene hubs were most similar to gene signatures from tongue basal cells, further emphasizing the importance of tissue context on the modes of CS. However, they were similar to other hubs from multiple tissues and cell types (Fig. 3E), either presenting as type-A or type-B senescence hubs but not both, suggesting this multimodal pattern is not restricted to tongue keratinocytes. Next, we used *SenePy* to score tongue keratinocytes based on these type-A and type-B signatures. The number of cells that were outliers ($> \mu + 3\sigma$) in their signature-specific *SenePy* score distributions increased significantly as a function of organismal age (hub 0,1 Chi-Square $p$-value = $7.9 \times 10^{-86}$, $7.9 \times 10^{-90}$) (Fig. 3F, G), indicating a higher number of senescent keratinocytes in old mouse tongues compared to young.

We next examined senescent fibroblasts from three different tissues to determine the CS characteristics of similar cell types in different contexts. The mouse fibroblasts from hearts, lungs, and tracheas each had two distinct CS signature hubs. Most fibroblast hub signatures shared little genetic similarity and high cosine distance (Fig. 3H). The senescent cell gene hubs with the highest pairwise similarities were present in the cells of the lungs and trachea, possibly indicating that the spatial proximity and function in the respiratory system may have resulted in similar CS phenotypes. Functionally, these similar hubs in the lungs and trachea shared common biological processes, such as inflammatory response, cytokine signaling, and immune cell chemotaxis (Fig. 3J). However, the trachea hub was uniquely and highly enriched for genes involved in B-cell signaling and neutrophil activation. When the fibroblast populations were scored with *SenePy*, they showed distinct temporal kinetics (Fig. 3I). In all cell populations, there was a small proportion of senescent cells in young mice and a drastic increase in old mice. The biggest increase in the proportion of senescent cells occurred between 18 and 24 months. Interestingly, cells identified using the most similar trachea and lung hubs had comparable temporal kinetics and nearly identical high proportions of senescent cells in 24-month-old mice. Senescent fibroblast populations in the heart and lungs also followed parallel kinetics despite greater gene and ontological distance. Together, these results suggest multiple modes of CS even within the same populations and that these modes are temporally and phenotypically distinct.

## Cell-specific signatures are unique but share common stress response and inflammatory pathways

We explored the CS markers we derived for gene and ontological commonality to determine whether there are universal CS signatures. To account for cell-type sampling bias and batch effect, we created a statistical model to determine which genes are overrepresented in the signatures without requiring their presence in every signature. No gene was present in every signature, but many genes were statistically overrepresented between all signatures compared to what is expected by random chance (Fig. 4A). Out of 9645 signature union genes, we identified a panel of 635 universal genes that were statistically overrepresented in the signatures (FDR $p$-value < 0.01) (Supplementary

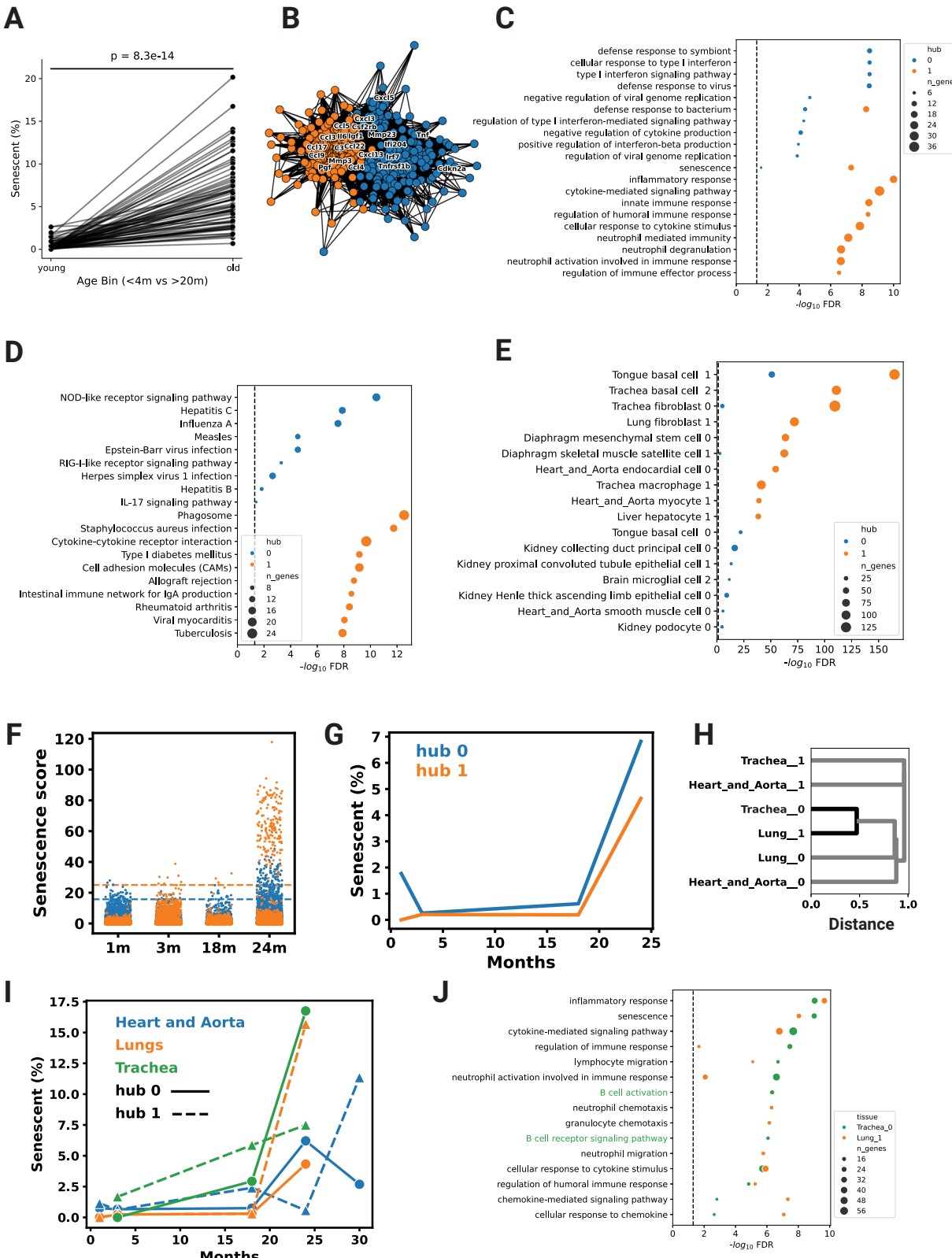

Data 7). There were 29 genes with greater than 25% prevalence ($n >= 12$, FDR $p$-value $\leq 3.1 \times 10^{-10}$) in the 43 mouse signatures. The most common gene, and the only one above 50% prevalence, was *Hba-a1*, which was present in 28 (60%, FDR $p$-value $< 8.6 \times 10^{-13}$) of the 43 signatures. For comparison, *Cdkn2a* (gene that encodes p16) was also overrepresented in the data but only present in 12 signatures (FDR $p$-value $= 3.1 \times 10^{-10}$). Among other known CS markers, multiple Bcl2

family genes, which are common senolytic targets, were over-represented in these data. Interestingly, along with *Hba-a1*, the hemoglobin subunits *Hba-a2*, *Hbb-b1*, and *Hbb-b2* were among these most representative genes, despite the absence of erythrocytes in the upstream analysis. We next tested if cell types that are found in multiple tissues, such as endothelial cells, fibroblasts, or macrophages, have their own cell-type-specific signature. There were 14 genes found

**Fig. 3 | Multiple modes of senescence exist within the same cell populations.**
**A** The proportion of cells from young (< 4 month) and old mice (> 20 month) expressing their respective *SenePy* signatures (Wilcoxon signed-rank test, one-tailed, $n = 72$ pairs). **B** Signature derived from mouse tongue keratinocytes. Each node represents a gene and the connections represent co-positivity in cells. Connections are weighted by Pearson's R. Nodes are colored by Louvain-based assignment to distinct hub signatures. **C** GO and (**D**) KEGG gene set enrichment of the two keratinocytes hub signatures. The "senescence" gene set is the pre-defined senescence marker used in this study. The vertical dashed line represents FDR $p = 0.05$. **E** Pairwise enrichment of the two keratinocyte hubs against all other

*SenePy* senescence signatures. **F** The strip plot depicts the score of each keratinocyte determined by *SenePy* using the aforementioned hubs. Horizontal dashed lines represent three standard deviations above the mean. **G** Temporal kinetics of the proportion of cells scored three standard deviations above the mean by *SenePy* for the two keratinocyte hubs. **H** Hierarchical clustering of fibroblast hub signatures from mouse lungs, tracheas, and hearts based on cosine similarity. **I** Temporal kinetics of the proportion of lung, trachea, and heart fibroblast cells scored high by *SenePy* using their respective signatures. **J** GO gene set enrichment of the most similar trachea and lung fibroblast hubs. Pathways specific to the trachea hub are colored green. All enrichment plots use BH-corrected Fisher's Exact *P*-values.

in every fibroblast signature (FDR $p$-value $= 2.3 \times 10^{-14}$). No gene was found in every endothelial signature, but Cdkn2a and other genes were statistically overrepresented (FDR $p$-value $= 9.2 \times 10^{-8}$). Likewise, there were genes overrepresented in the macrophage signatures (FDR $p$-value $= 1.1 \times 10^{-7}$), but no genes were found in every macrophage signature (Supplementary Data 8).

Genes overrepresented in every mouse signature (FDR $p$-value < 0.05) were enriched for multiple biological processes involved in inflammation, immunity, cytokine signaling, and chemotaxis (Fig. 4B). The NF-kappa B signaling pathway, a known driver of cell senescence, was among the enriched pathways (KEGG, BH-corrected $p$-value $= 1.3 \times 10^{-5}$), emphasizing that NF-kappa B plays an important role in some programs of in vivo CS. However, only 9 of the 43 signatures were individually enriched for NF-kappa B signaling, indicating that it is far from universal (Fig. 4C). Therefore, to test for transcription factors that might be active, we tested our signatures for TF binding enrichment in their gene promoters (Fig. 4D). The most universally enriched binding motif was RREB1, which was enriched in 38 of 43 signatures. In addition, we found 46 other transcription factors enriched in over half of the mouse signatures.

The universal mouse *SenePy* signature had no overlap with the 330 global mouse aging genes identified from *Tabula Muris Senis* by Zhang et al. (Hypergeometric $p$-value $= 2.3 \times 10^{-5}$) (Supplementary Fig. 2f). However, the universal mouse gene signature does share significant overlap with known markers of cellular senescence (Hypergeometric $p$-value $= 1.4 \times 10^{-13}$). Conversely, the global aging signature derived from differential expression does not (Supplementary Fig. 2e). This supports our earlier observation that cellular senescence is distinct from organismal aging.

In human cells, only three genes were present in > 25% of the human CS signatures (Fig. 4E): matrix metallopeptidase 9 (*MMP9*), Myosin light chain 9 (*MYL9*), and Integral membrane protein 2 C (*ITM2C*) (FDR $p$-values $8.1 \times 10^{-8}$, $5.0 \times 10^{-7}$, $5.0 \times 10^{-7}$). *MMP9* is a known SASP component and is present in our curated set of CS markers. In comparison, *CDKN2A* was only present in 9 of 45 signatures, which is still higher than what would be expected by random chance (FDR $p$-value $= 3.4 \times 10^{-4}$). There were 734 genes overrepresented (FDR $p$-value < 0.01) in the human cell-type *SenePy* signatures (Supplementary Data 9). Genes in this universal signature were enriched for innate immune and other biological pathways (Fig. 4F). When signatures were tested individually, the most commonly enriched pathways included neutrophil-mediated immunity, platelet degranulation, cytokine signaling pathways, and other inflammation and immune pathways (Fig. 4G), consistent with SASP.

The most universal genes and pathways active in the cell-type signatures from both species shared some characteristics. There were 46 common genes that were enriched in both the mouse and human cell-type signatures. This was only a marginal over-representation compared to random chance (Hypergeometric, $p$-value $= 0.09$), suggesting low gene-wise concordance between species. *CDKN2A, CXCR2*, and *CCL3* were the only common genes that were previously established CS markers. However, the pathway concordance between species was high and both sets of common genes were enriched for 32 common pathways, such as

NF-kappa B signaling, AGE-RAGE signaling, and chemokine signaling. Likewise, eight of the top 20 most commonly enriched transcription factors from mouse signatures were also commonly enriched in human signatures (Fig. 4H). This suggests that core CS pathways between species are conserved but the individual genes that are enriched in senescent cells show a high degree of genetic variation. Together, these data indicate that our de novo cell-type signatures are enriched for known CS phenotypes and share some commonality between cell types and species despite their high degree of heterogeneity.

## The cell-specific kinetics of senescent cell accumulation with organismal age

We next used *SenePy* to determine the proportion of senescent cells in distinct populations from young and old organisms. The numbers and proportions of cells identified as senescent increased drastically with age in both mice (Fig. 5A) and humans (Fig. 5B) overall but showed distinct cellular tropism and kinetics. Within the tested mouse cells, pancreatic acinar cells had the highest proportion of senescent cells at old age, followed by tracheal and lung fibroblasts (Fig. 5C). Out of the 45 tested cell populations in humans with appreciable CS signatures, skin hair follicle cells had the highest percentage of senescent cells at the relatively young age bin of 48–58. Lung type II pneumocytes had the second highest percentage of senescent cells, reaching a level of greater than 11% in patients aged 68–77 (Fig. 5D). However, in the human cells, we observed temporal heterogeneity between patients. For example, cardiomyocytes from a patient in the 58-year bin were 7.6% senescent while a patient in the 68-year bin had only 0.4% senescent cardiomyocytes. This disparate pattern of fewer senescent heart cells in the older patient persisted in every other heart cell type except adipocytes, suggesting patient-specific differences are important drivers of CS within tissues globally. In younger individuals (ages 40–60), the heart cell type with the highest proportion of senescent cells was invariably lymphoid cells (Fig. 5E). Conversely, there was a large increase in the proportion of senescent cells within solid tissue cell types taken from most 60-year + donors. One 68-year-old patient, in particular, had a surprisingly high proportion of cardiomyocytes and fibroblasts with high CS scores. However, in the oldest heart test, there was a relatively low proportion of senescent cells in all cell types except for lymphocytes.

The proportions of cells within populations predicted to be senescent by *SenePy* were not correlated to the replicative potential in their respective populations (Supplementary Fig. 5a, b). Surprisingly, replicative populations, such as large intestine enterocytes, had minimal increases in the proportion of senescent cells with age. This is corroborated by an undetectable change in the number of *Cdkn2a +* enterocytes with age and the general lack of correlation between *Ckdn2a +* cells in replicative populations. Likewise, we did not observe a negative correlation between the population-level expression of telomerase and the calculated senescence burden (Supplementary Fig. 5c, d). Thus, suggesting that the well-known driver of in vitro CS, telomere attrition, may not be the primary driver of CS within organisms.

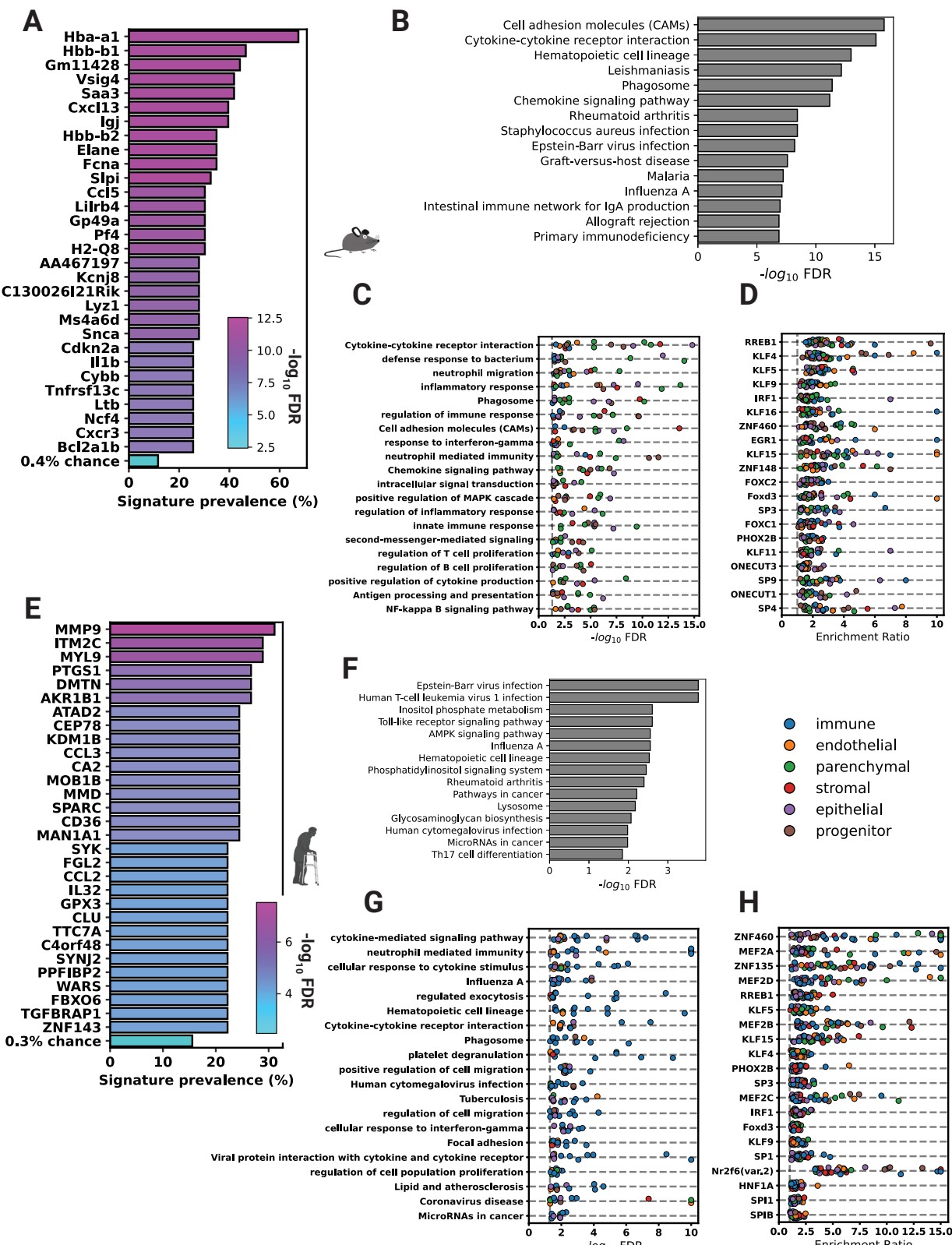

**Fig. 4 | Cell-specific signatures are unique but share some genes and biological pathways. A** Plot depicting the most commonly found genes from the novel mouse cell-specific signatures. Significance depicts how likely those genes would be found in that many signatures by random chance. **B** The 15 most enriched KEGG pathways from the universal *SenePy* signature. **C** The most commonly enriched KEGG and GO gene sets in every *SenePY* mouse signature. **D** The most commonly enriched transcription factor motifs in the promotors of *SenePy* mouse signature genes. **E** Plots depicting the most commonly found genes from the *SenePy* human cell-specific signatures. **F** The 12 most enriched KEGG pathways from the universal *SenePy* human signature. **G** The most commonly enriched KEGG and GO gene sets in every human *SenePy* signature. The bars note the percent of signatures enriched for the given pathway. **H** The most commonly enriched transcription factor motifs in the promotors of human *SenePy* signature genes. The human icon was created in BioRender. Sanborn, M. (2025) https://BioRender.com/l04t362.

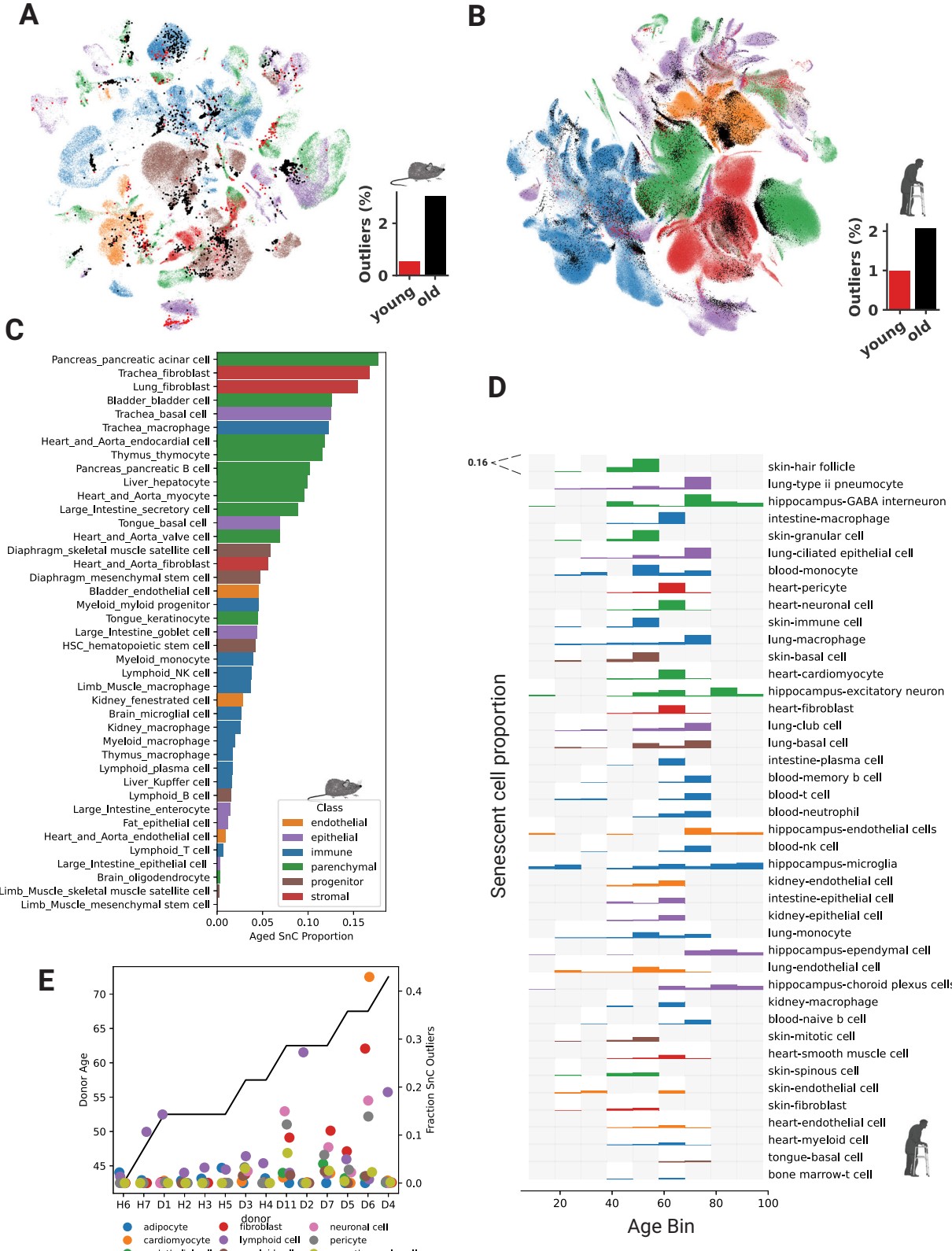

Fig. 5 | The cell-specific kinetics of senescent cell accumulation with organismal age. UMAPs of (**A**) mouse and (**B**) human cells depicting broad cell classification and overlayed with cells that were outliers determined by their *SenePy* score. **C** The proportional increase of *SenePy* outlier cells in old mice (24- or 30-month) relative to 3-month-old mice. **D** The proportion of *SenePy* human outlier cells across age bins. Each row represents cell proportions from 0–0.16 and gray rectangles note that no data is available. **E** The fraction of *SenPy* outliers in individual heart tissue cells stratified by the donor. Age increases along the *x*-axis from left to right. The human icon was created in BioRender. Sanborn, M. (2025) https://BioRender.com/l04t362.

### *SenePy* identifies ground-truth in vivo cellular senescence more robustly than established markers

To test *SenePy*'s ability to robustly detect senescent cells in additional datasets, we utilized single-cell RNA-seq data from p16-Cre[ERE2]-tdTomato reporter mice in which p16 + cells become red fluorescent[32]. We first performed differential expression analysis between the tdTomato + and tdTomato- kidney cells (Fig. 6A). The mRNAs differentially abundant in tdTomato + cells were enriched for kidney epithelial signatures derived in this study. Our kidney epithelial signature was more enriched than previously defined gene sets or a CS gene panel derived from the AI model GPT-4 that was tasked to identify CS signatures (*n* = 100 genes, "senGPT") (Fig. 6B). A notable shift was observed in the distribution of cells scored with *SenePy* using kidney-specific signatures but not in a heart endothelial cell hub (Fig. 6C). Likewise, we analyzed *senePy* liver signatures and found mRNAs more abundant in tdTomato + liver cells were more enriched for *SenePy*'s liver Kupffer cell signature than any other gene sets (Supplementary Fig. 6a). These data indicate that *SenePy*'s CS signatures and derived scores identify cells that become p16 +.

We also examined the ability of *SenePy* signatures to identify transcriptional changes due to senolytic treatment as well as those seen in experimental conditions that induce CS in multiple in vivo and in vitro models. The genes downregulated in mouse lungs following therapeutic senolysis were enriched for multiple *SenePy* lung- or airway-specific signatures (Fig. 6D). *SenePy* signatures were more enriched than cell-type agnostic gene sets. *SenePy* does not contain a specific skeletal muscle CS signature due to data availability, yet mRNA less abundant in mouse muscle tissue following senolytic treatment was enriched for multiple *SenePy* signatures, including one from myocytes (Supplementary Fig. 6d). Multiple *SenePy* endothelial signatures were enriched in mRNA more abundant after radiation-induced CS of human endothelial cells, but the advantage of *SenePy* over other gene sets was diminished in this in vitro context (Supplementary Fig. 6b, e). The discrepancy between the in vitro and in vivo efficacy of *SenePy* was even more apparent in models of senescent fibroblasts in culture (Supplementary Fig. 6c and Fig. 6E). In these in vitro contexts, *SenePy* was outperformed. These data suggest that *SenePy* recapitulates in vivo cellular senescence and that gene sets derived primarily through previous in vitro experiments do not.

Next, we tested the marker suitability of genes that encode for p16[ink4a] and p21[cip1] in the p16-Cre[ERE2]-tdTomato liver cells (Supplementary Fig. 6f). Only 8% of p16[high] cells, as indicated by tdTomato, had detectable levels of *Cdkn2a* RNA (p16 encoding gene) (Supplementary Fig. 6g). This suggests that either *Cdkn2a* expression was not detectable due to single-cell dropout or the expression of *Cdkn2a* is transitory in the majority of these senescent cells. Another important marker of cellular senescence, *Cdkn1a* (p21 encoding gene), was found in the majority of both tdTomato+ and tdTomato- cells, making its binary expression an inadequate metric because it's more universally expressed in non-senescent cells (Supplementary Fig. 6h, i). These data represent a striking example of why sole reliance on known cellular senescence genes like p16 and p21 is not sufficient, especially in single-cell transcriptomics because of low single-cell resolution, dropout, and marker-independent CS programs.

For additional in vivo validation, we analyzed flow-sorted senescent hepatocyte RNA data from a mouse model of oncogene-induced senescence[33]. Chan et al. induced cellular senescence in mouse hepatocytes using NRAS expression and harvested hepatocytes during peak senescence at 12 and 30 days, along with tumor and healthy tissue at 218 days (Fig. 6F). The cells follow multiple CS pseudotime trajectories from the mV (mVenus) control root (Fig. 6G). The senePy mouse hepatocyte signature is composed of two distinct hubs (hepatocyte 0 and hepatocyte 1). Both hubs strongly correlate to distinct pseudotime trajectories (Pearson's *p*-value = 2 × 10[−232], *p*-value = 0), supporting the idea that a single cell type can have multiple CS phenotypes (Fig. 6H).

The universal senePy signature score is significantly correlated to global pseudotime (*p*-value = 0) (Fig. 6I). Cells scored using traditional CS markers and the senMayo gene set were not as strongly associated with CS pseudotime (Fig. 6J). The *SenePy* score was inversely correlated to the hepatocyte marker, *Albumin*, likely due to reduced cell identity associated with cellular senescence (Fig. 6K). Independent of pseudotime, senePy scores were higher in senescent hepatocytes relative to the control than cells scored with traditional CS markers (Fig. 6I). There was a 4.3-fold increase in the mean senePy score at day 12 (Mann Whitney, one-tailed, *p*-value = 2.9 × 10[−145]) and a 4.1-fold increase at day 30 (*p*-value = 3.7 × 10[−177]) relative to the control. Conversely, CS marker-based scoring did not increase at day 12 and had a 1.3-fold increase at day 30 (*p*-value = 1.6 × 10[−17]). These results indicate that senePy cell-specific signatures and its derived universal signature robustly recapitulate in vivo CS within mouse hepatocytes.

We also tested an additional scoring method, scDRS[34], which is capable of handling the network centrality weights central to *SenePy* signatures. Both the default binary and the normalized methods of *SenePy* were highly correlated to scRDS scores (Pearson's R, *p*-value = 0) (Supplementary Fig. 7a). The cells identified by scDRS to have an FDR p-value less than 0.05 had very significant overlap with every *SenePy* outlier threshold tested (Hypergeometric, *p* < 1.9 × 10[−116]) (Supplementary Fig. 7b). Cells identified by higher *SenePy* thresholds had high overlap with scDRS, but the overall Jaccard Index decreases as the number of *SenePy* cells decreases relative to scDRS cells (Supplementary Fig. 7c). This indicates that higher thresholds may be more specific for senescent cells but with less sensitivity. Both the default binary *SenePy* method and its normalized count method had high overlap with each other and scDRS (Supplementary Fig. 7d). However, the scores produced by the default binary method produce multi-modal distributions, which make it simpler to empirically derive a threshold (Supplementary Fig. 7c, f). All three methods detected a similar increase in the number of senescent cells with age (Supplementary Fig. 7e). When tested in the p16 reporter cells, all three methods generated significantly higher scores in the tdTomato + cells compared to tdTomato- cells (Supplementary Fig. 7f). The scDRS method is incorporated into the *SenePy* package as an additional scoring flavor.

### *SenePy* predicts an elevated cellular senescence burden in severe disease

We posited that *SenePy* could be applied to datasets to quantify the burden of CS in disease. We first used *SenePy* to analyze a single-cell RNAseq lung dataset of 19 individuals who died from COVID-19 and 7 age-matched controls[35] (Fig. 7A). There were senescent lung cells in both control and COVID-19 patients, but there was an observable increase in senescent lung epithelial cells among COVID-19 patients (Fig. 7B). COVID-19 mortality was significantly associated with an increased proportion of senescent lung epithelial cells and were present in AT1, AT2, and general airway epithelial cell populations (Mann-Whitney, two-tailed, *p*-value = 0.004) (Fig. 7C). The proportions of CS in non-epithelial cells, such as immune cells, fibroblasts, and endothelial cells, were not significantly associated with COVID-19 mortality (Fig. 7D). Some of the deceased patients had relatively elevated senescent cell burdens in multiple cell types (Fig. 7E).

Next, we used *SenePy* to score spatiotemporally resolved mouse transcriptomics data following myocardial infarction[36]. We used the hub signatures we previously derived from mouse hearts to score the spatially resolved spots (Fig. 7F). Senescent loci were found even in the control heart, corroborating earlier observations that even young organisms have baseline levels of cellular senescence. However, since the spatially resolved spots consist of multiple single cells, the number of single senescent cells in these data are unknowable. The proportion of spots with high CS burden was highest at day 7 but the change was not statistically significant, likely due to the small sample size (ANOVA,

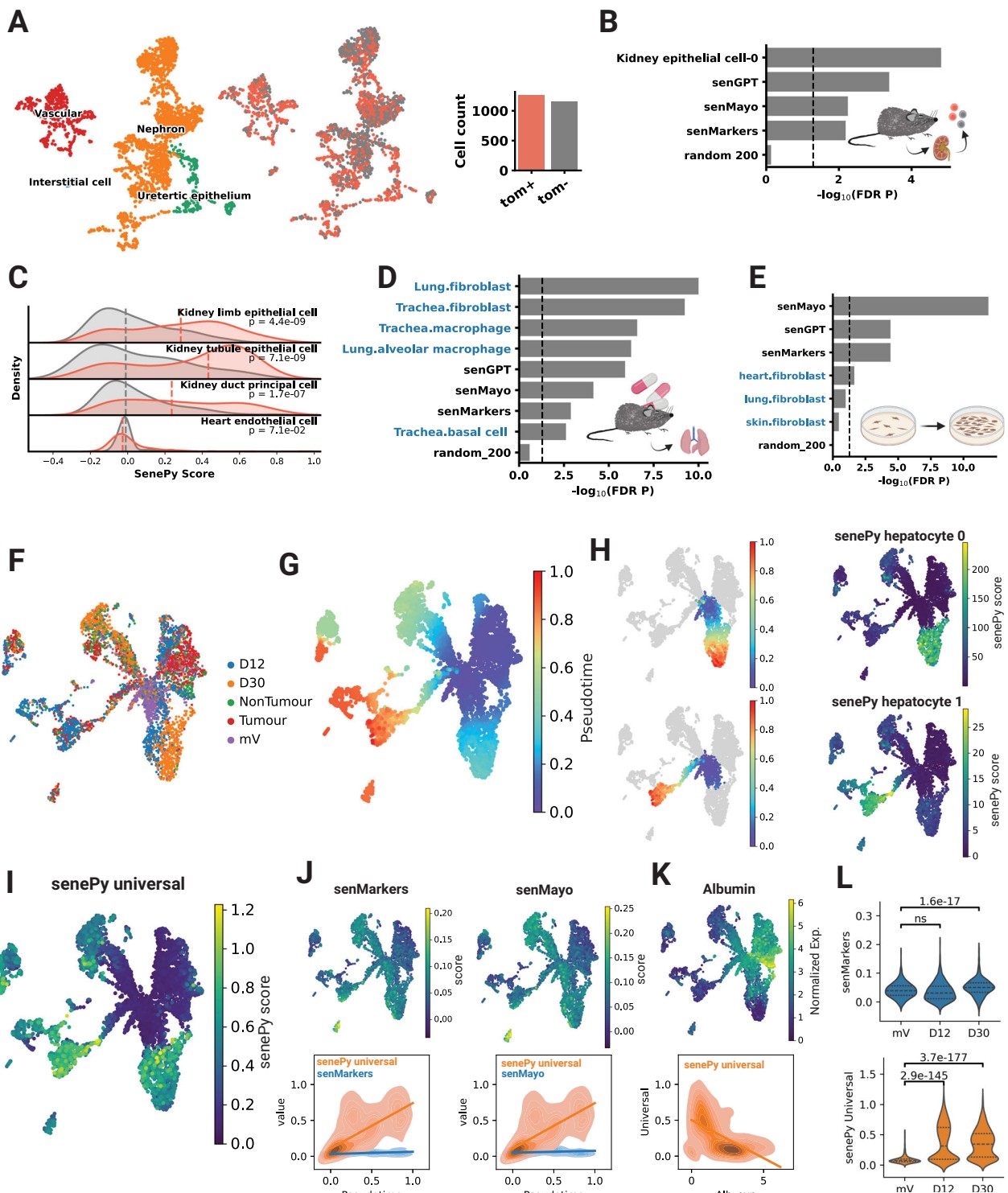

**Fig. 6 | SenePy identifies ground-truth in vivo cellular senescence more robustly than established markers. A** UMAPs of single-cells from the kidney which were enriched for td-Tomato+ cells. **B** Enrichment analysis of differentially abundant gene mRNA in the td-Tomato + kidney cells. Gene sets were derived in this study and we also include the SenMayo signature, CS markers from the literature (senMarkers), and a CS gene set derived from Chat-GPT4 (senGPT). Only *SenePy* kidney epithelial signatures were tested (Fisher's Exact FDR *p*-value). **C** Density plots depicting *SenePy* score distributions calculated in kidney cells using kidney-specific *SenePy* signatures. Each row label depicts the senescence signature used. The color indicates if the cells were td-Tomato + (Mann Whitney, two-tailed). **D** Enrichment analysis of lung tissue genes downregulated after senolytic treatment in mice. Blue labels indicate *SenePy* signatures (Fisher's Exact FDR *p*-value). **E** Enrichment analysis of gene mRNAs more abundant in replicative fibroblast

senescence (Fisher's Exact FDR *p*-value). **F** UMAP of single mouse hepatocytes undergoing KRAS oncogene-induced senescence (Chan et al. 2024). mV: mVenus control, D12: day 12, D30: day 30. Tumor and non-tumor were harvested at 218 days after KRAS induction. **G** Pseudotime projected onto the hepatocytes with a mVenus root cell. **H** Two distinct trajectories of pseudotime are closely associated with the scores from both *SenePy* hepatocyte signatures. **I** Cells scored using the *SenePy* universal mouse signature. **J** Cells scored with either senMarkers or senMayo (top). Regression and density plots depicting the association between normalized signature scores and pseudotime (bottom). **K** Association between the *SenePy* universal signature score and the expression of Albumin. **L** Scoring based on literature marker genes and the universal *SenePy* signature (Mann-Whitney, one-sided). The image icons were created in BioRender. Sanborn, M. (2025) https://BioRender.com/l04t362.

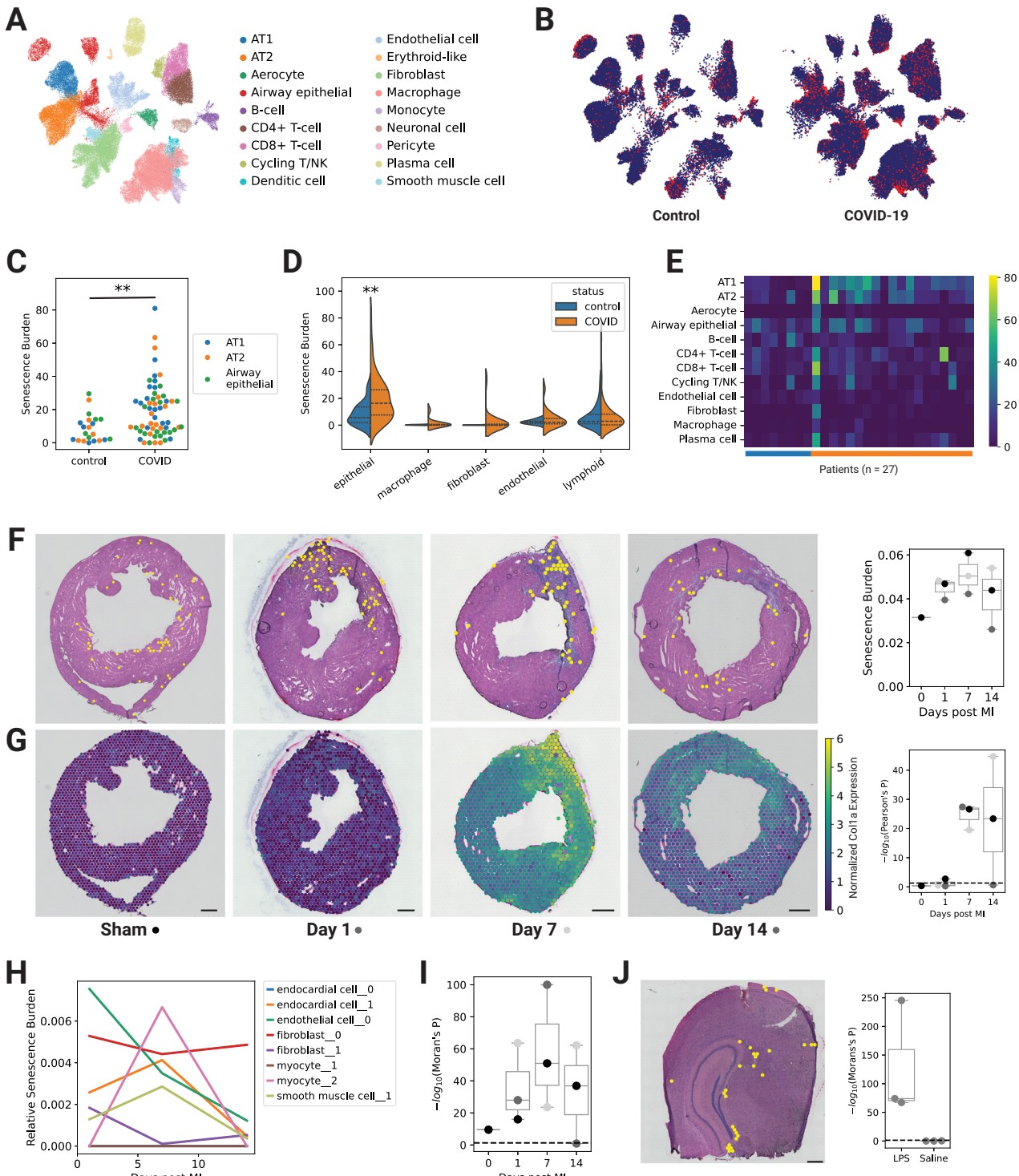

**Fig. 7 | *SenePy* predicts elevated senescence burden in severe disease. A** UMAP showing cells from uninfected control lungs (*n* = 7) and patients who died from COVID-19 (*n* = 20). **B** UMAP of control cells (left) and COVID-19 cells (right) with *SenePy* outlier cells labeled. **C** Proportion of cells identified as outliers with *SenePy* from AT1, AT2, and airway epithelial cells (*p* = 0.004, Mann-Whitney, two-sided). **D** Distributions of *SenePy* score from the major cell lung cell classes (Mann-Whitney, two-sided, ** *p* = 0.004). **E** Heatmap with the relative senescence burden of each cell type in each patient. **F** Representative whole heart H&E staining overlayed with spatially resolved 10x Visium spots. Yellow spots are identified as senescence outliers from their *SenePy* score. Box plot (right) shows the proportion of identified spots at each time point (*p* = 0.30, one-way ANOVA, individual replicate points shown). **G** The bottom images represent post-MI fibrosis via normalized *Col1a1*

expression. Scale bars represent 500 μM. The correlation between senescence-like spots and *Col1a1* expression for each sample is shown by the right box plot (− log$_{10}$[Pearson's R *p*-value, two-sided]). The indicated gray-scale dots match the shown images to their respective data points in the box plots. **H** The relative contribution to the overall calculated senescence burden from the 8 hubs used to score the spots. **I** Spatial autocorrelation of the senescence-like spots (− log$_{10}$[Moran's I *p*-value]). The horizontal line represents *p* = 0.05. **J** Representative H&E image of coronal sections spatially resolved by 10x Visium which were taken from mice 24 h after exposure to LPS. Yellow spots are identified as senescence outliers from their *SenePy* score. The summary plot (right) depicts spatial autocorrelation of spots in LPS and saline-treated mice (− log$_{10}$[Moran's I *p*-value], *n* = 3). Box plots in Fig. 7 depict the median and full range of values, with individual points shown.

$n \leq 3$). We observed a strong spatial association between spots with high CS burden and heart fibrosis after infarction (Fig. 7G). Senescent-like spots strongly colocalized in regions of the hearts expressing fibrotic markers such as *Col1a1*, which is not a gene in the *SenePy* signature. This correlation becomes readily apparent by 7 days post-MI but was not observed in the control heart or hearts shortly after MI. The 8 hubs used to score the spots contributed to the overall CS burden with distinct temporal patterns (Fig. 7H). The endothelial cell hub had the highest contribution at day one but continued to decrease up to day 14. The score from a myocyte hub jumped from baseline on day 7 then dropped back down. Other CS gene programs remained otherwise unperturbed by MI.

In addition, we observed strong spatial autocorrelation between the spatially-resolved spots with high CS burden (Fig. 7F, I). Only one of the 9 samples did not have highly significant spatial clustering of senescent-like spots. Unsurprisingly, this same day-14 sample has low senescent spot association to *Col1a1* as well as the lowest overall CS burden. To further investigate this finding of senescent cell clustering and to see if this phenomenon is apparent in other tissues, we utilized spatial transcriptomics data from mouse brains before and after inflammatory insult (Fig. 7J). The control brains had a small number of senescent-like spots. However, the brains from LPS-treated animals had amplified CS signatures and highly significant clustering of senescent-like spots. These data indicate that senescent cells are more likely to be found in close proximity across multiple in vivo systems.

## Discussion

There is a paucity of tissue- and cell-specific markers for senescent cells due to the heterogeneity of cells that undergo cellular senescence (CS). This is especially challenging in single-cell transcriptomics because the high rate of dropout and limited sequencing depth of the technology poses a challenge for using classical CS markers such as *Cdkn2a* and *Cdkn1a* as sole indicators of cell senescence. Furthermore, the cell-specific heterogeneity of CS represents a major challenge for the development of a universal CS signature gene panel. Therefore, in this study, we took an unbiased large-data approach to identify cell-specific programs of cellular senescence and created *SenePy* as an open-source platform (https://github.com/jaleesr/SenePy) to identify senescent cells in single-cell transcriptomic data. We validated the *SenePy* approach using single-cell RNA-seq data in multiple models and applied *SenePy* to determine the kinetics and heterogeneity of CS across several human and mouse cell types in aging and disease.

Previous studies have generated transcriptomic signatures of CS based on induced CS in controlled in vitro environments[13,20]. While these signatures have helped advance the mechanistic understanding of CS, it is challenging to use such signatures for highly variable in vivo contexts. We show that cells express many of these genes at higher rates with age at the organism or tissue level, but none of the genes obtained from such CS panels were applicable in the majority of cell types tested. Even the widely used marker *CDKN2A* was identified in senescent cells in less than a third of mouse and human cell types. Our analysis of the p16-Cre$^{ERE2}$-tdTomato mouse cells highlights the limits of using a small number of markers, such as p16$^{ink4a}$ or *Cdkn2a* in transcriptomics data. While the tdTomato+ cells likely represented bona fide senescent cells[32], only a small proportion of individual cells had detectable levels of *Cdkn2a* at the time of tissue harvest and sequencing. This likely arose from the transient expression of p16 earlier on in the CS program in combination with gene dropout inherent in single-cell sequencing. Nevertheless, many highly visible and impactful studies are forced to rely on a small, unspecific set of markers because better alternatives did not exist.

Our approach focused on CS markers specific to individual cell populations that would not be confounded by transcriptional differences that are merely reflective of organismal aging. We were able to extract cellular sub-population level programs of CS by setting kinetic thresholds based on the prior knowledge that senescent cells increase with age but are present in the minority of cells[31] and by examining single-cell co-expression as opposed to differential expression. Unsurprisingly, many of our computationally-derived signatures contained and were statistically enriched for pre-established CS markers. However, these comprised the minority and most genes we identified have not been considered as CS markers. While we did find well-known CS genes like *CDKN2A*, *BCL2*-family genes, and various SASP factors to be statistically common within our signatures, the most universal markers were novel ones. For example, in mice, the most common signature gene we identified was the alpha hemoglobin subunit. Hemoglobin has been previously reported as a response gene to oxidative stress in non-erythrocytes[37,38] but has yet to be reported in CS. Interestingly, in human signatures, there was no significant over-representation of globin genes. We observed this and other important differences between mouse and human signatures, suggesting organism-specific CS marker panels to be more specific. Some of the shared pathways and genes between signatures share commonality with recent single-cell analysis of senescent immune cells[39]. Both studies show an upregulation of NF-kappa B signaling, BCL2, and antigen presentation and processing, and our results show that this is a common feature in many cell types.

We used our signatures to map the kinetics of senescent cell accumulation in many different tissues and cell types. Recent work has mapped the abundance of senescent marker mRNA in 13 different tissues as a function of mouse age and in a progeria model[40]. This work, however, was agnostic of cell type and relied on a small set of CS markers. Our study comprehensively maps the increase in senescent cells in many different mouse and human tissues with respect to cell type. Furthermore, we used *SenePy* to examine trajectories of CS during hepatocyte tumorigenesis. *SenePy* identified multiple trajectories of CS with distinct phenotypes, consistent with the biological findings in the original study[33]. The universal *SenePy* signature strongly correlated to CS development in this hepatocyte-specific context. These data serve as an additional validation of *SenePy* and show that it can be used to study the process and kinetics of CS induction in disease settings.

Senescent cells contribute to cardiovascular pathology[6,41], but their role in disease has never been spatiotemporally characterized. We show that senescent cells localize at sites of heart fibrosis. The proportion of senescent cells in each heart throughout the time series did not significantly change and were present even in the control heart. We also observed highly significant spatial clustering of senescent foci in the hearts and brains of mice which may be supporting evidence for an in vivo bystander effect[15]. The spatial distributions of senescent cells have been previously examined from spatially resolved transcriptomics data in aged mouse brains[42]. Their results show that *Cdkn2a* + spots are adjacent to activated microglia but show no spatial clustering of *Cdkn2a* + spots. However, their methodology relies on a narrow definition of CS, which may not translate to actual p16$^{ink4a}$ and does not account for Cdkn2a dropout or p16-independent forms of CS. These and other data would greatly benefit from a reexamination with more comprehensive gene sets, such as those proposed herein.

By design, our methodology removes genes that are constitutively expressed at baseline or in aged cells to maintain a distinction from organismal aging. Inherently, this discounts genes that may be part of the CS program which overlap with the transcriptional shift with age. We also do not account for genes that are down-regulated in senescent cells. Negative markers of CS would add extra information to better identify senescent cells, but to find negative correlations in all pairwise combinations of genes with this methodology was computationally limiting. The comprehensiveness of our signature panel is also limited by the data available at the time of study and the exact set of tissues and cells tested influences any conclusions of universality or comparison between species. Single-cell noise and

batch effect may also obfuscate a complete universal CS signature. We do not expect our signatures to negate the need for large-scale future efforts such as SenNet[21]. We instead expect our work to be complementary and assist these efforts. This work may also serve as a starting point for studying how patient-specific factors such as sex and lifestyle impact distinct senescent cell phenotypes and the kinetics of senescent cell accumulation.

This work comprehensively identified gene expression programs and signatures of senescent cells that are stratified by species, tissue, and cell type and used them to broadly characterize senescent cells in mice and humans. We created *SenePy*: a computational platform that assigns a CS score to individual cells in single-cell transcriptomic data, which can serve as a resource to uncover cell-type and tissue-specific mechanisms of cellular CS in vivo.

## Methods

### Data collection
Single-cell RNA mouse data were collected from the *Tabula Muris Senis* atlas[22]. *Tabula Muris Senis* consists of single cells from 30 mice from 1 to 30 months of age taken from 19 tissues. Human single-cell data were collected from 7 studies. The liver data were obtained from five donors ranging from 21 to 65 years old[29]. Single skin cells were obtained from 6 patients ranging in age from 18 to 48 years old[25]. Lung data were collected from 17 donors ranging from 21 to 72 years old[28]. Human heart cells were taken from 14 patients ranging from 40 to 75 years old[26]. Human hippocampal cells were collected from 37 patients ranging from newborn to 92 years old[23]. These tissue-specific datasets were given priority for downstream analysis in their respective tissues, but we used additional multi-tissue atlases. Cells from the *Human Cell Landscape* came from 51 donors ranging from 21 to 66 years old and from 25 different tissues[27]. Cells from *Tabula Sapiens* came from 15 patients ranging from 22 to 74 years of age[24].

### Data annotation
The data were available in a range of formats from fastq to processed and annotated count data. Data from *Tabula Muris Senis*, *Tabula Sapiens*, and the human heart study were provided with cell type annotations. The Lung and hippocampal studies provided unannotated counts. Fastq data were processed through 10x CellRanger or the Dropseq protocol (https://mccarrolllab.org/dropseq/) depending on the technology used to prepare the libraries. All human fastqs were aligned to GRCh38. Processed single-cell counts were handled with Scanpy[43]. Cells were filtered out if they had a relatively low or high number of detected genes or a high relative proportion of mitochondrial reads (thresholds varied based on dataset distribution). We used a variety of methods to annotate cell types. Since the *Human Cell Landscape* contained many tissues and cell types, we transferred the annotations from *Tabula Sapiens* using scANVI[44] after processing the raw data with scVI-tools[45]. If a cluster, which was called by the Leiden algorithm on the scVI embeddings, had lower than 85% cell-type agreement, those cells were not used in downstream analysis. Cell types from the liver, skin, and lung studies were annotated similarly but clusters with poor label transfer were instead manually annotated using known cell-type markers[46]. For lack of a reference dataset, the hippocampal data were annotated exclusively using known markers. Annotations were harmonized across datasets (e.g., "kidney endothelial cell" changed to "endothelial cell") and mapped back onto the raw counts. Cells lacking annotations, because they failed QC or label transfer were discarded. For total dataset visualizations, the species-specific raw data were integrated using scVI, and the embeddings were projected via UMAP.

### Mouse cell-type specific age-dynamic genes
Mouse data came from mice aged 1, 3, 18, 21, 24, and 30 months (m) but age availability varied by tissue. Cells were stratified by tissue, age,

and cell type. The starting baseline was chosen as 3 m if there were at least 200 3 m cells, if not the starting baseline was aggregated with 1 m cells. Likewise, 30 m was prioritized for old cells if at least 200 were present, otherwise the old baseline fell back to 24 m. The proportion of cells expressing one or more UMI copies of a gene was determined in each population (Eq. 1). Zero values at 3 m or 1 m were imputed with the inverse of the cell count ($p_{age} = n_{total\ cells}^{-1}$). Young cells were used as baselines and the ratios in old cells were determined relative to them (Eq. 2). For each cell type an age-dynamic score was calculated for each gene that is a sum of individual weighted metrics: young proportion, old proportion, gain, and ratio (Eq. 3). Each metric has an ideal range which roughly reflects the expected dynamics of senescent cells with age. We created a null distribution for each cell type by shuffling the gene by cell matrix for each cell type 1000 times, resulting in around 20 million null values for each cell type. The data were shuffled across cells to correctly model and account for the actual sparsity within each cell type. To determine if a gene was significantly dynamic, its observed value was compared to the null distribution. Multiple slopes and parameters were tested for the weighted metric functions but the resulting comparison between observed and null values was very stable. Dynamic genes are cell-specific markers and do not account for small changes to baseline levels of constitutively expressed genes, which may be senescence-associated genes but are not specific markers.

$$p_{age} = n_{gene^+\ cells} / n_{\text{total cells}} \tag{1}$$

(Eq. 1) $p_{age}$: the proportion of cells positive (i.e., expressing a gene) for a gene at a given age. Where $n_{gene^+\ cells}$ is the number of cells positive for a given gene and $n_{\text{total cells}}$ is the number of total cells in the same population.

$$r_{30m|24m} = p_{30m|24m} / p_{3m|1m} \tag{2}$$

(Eq. 2) $r_{30m|24m}$: ratio of old cells positive for a gene relative to young cells positive. Note that $p_{30m|24m}$ represents the proportion of cells positive for a gene in cells from 30- or 24-month-old mice and $p_{3m|1m}$ are the proportion of cells positive for the gene from 3- or 1-month mice.

$$old(x) = \begin{cases} \frac{1}{1+e^{-2x}} & \text{if } 0 < x \le 3 \\ 1 & \text{if } 3 < x \le 20 \\ -\frac{1}{4}x + 6 & \text{if } x > 20 \end{cases}$$

$$gain(x) = \begin{cases} \frac{x}{5} & \text{if } x < 5 \\ -\frac{x}{5} + 4 & \text{if } x > 15 \\ 1 & \text{otherwise} \end{cases}$$

$$young(x) = \begin{cases} 1 & \text{if } x < 5 \\ -\frac{x}{2} + 3.5 & \text{otherwise} \end{cases}$$

$$ratio(r_{30m|24m}) = \begin{cases} r_{30m|24m} & \text{if } r_{30m|24m} < 2.5 \\ 2.5 & \text{otherwise} \end{cases}$$

$$GAD = old(x) + gain(x) + young(x) + ratio(r_{30m|24m}) \tag{3}$$

(Eq. 3) *GAD*: An individual gene age-dynamic score is a sum of the weighted metrics. $x$ is the percent of cells positive for the given gene in old cells, young cells, or the difference between the two (gain).

## Human cell-type specific age-dynamic genes

Human ages were binned into 10-year bins to account for the continuous range of human ages. Bins were indexed from 8 (8–17 years old) to 88 (88–97 years old), with 9 total bins. To be considered for further analysis a cell-type population must 1) have three unique age bins with at least 100 cells in each bin or a bin with age ≤ 28 and a bin with age ≥ 58 with at least 100 cells and 2) have a bin with age ≥ 48 with at least 100 cells. These criteria were required in individual datasets to avoid confounding effects from multiple studies. Cells were stratified by dataset, tissue, age bin, and cell type. The proportions of cells expressing genes were calculated for each age bin (Eq. 1). The young starting populations were selected from the 8, 18, or 28 year bins if one was present, else the starting proportion was calculated by regressing the age and known proportion values and solving for 18 years (Eq. 4). The old ending populations were selected from the oldest age bin ($p_{old}$) in a given cell type.

Genes were considered dynamic if 1) the trend of age by proportion of cells expressing the gene was positive (Eq. 5) and the gene *GAD* (Eq. 6) was significant when compared to a null distribution.

$$\text{cov}(\text{age}, p) = \frac{\sum (\text{age}_i - \overline{\text{age}})(p_i - \bar{p})}{n}$$

$$\text{var}(\text{age}) = \frac{(\text{age}_i - \overline{\text{age}})^2}{n}$$

$$p_{18} = \frac{18 \cdot \text{cov}(\text{age}, p)}{\text{var}(\text{age})} + \bar{p} - \frac{\overline{\text{age}} \cdot \text{cov}(\text{age}, p)}{\text{var}(\text{age})} \tag{4}$$

(Eq. 4) $p_{18}$: extrapolated proportion of cells positive in the 18-year bin. Where $\text{cov}(\text{age}, p)$ represents the covariance between age and proportion $p$ (Eq. 1), $\text{var}(\text{age})$ is the variance of age, and $n$ is number of age bins represented in the data.

$$m = \frac{\text{cov}(\text{age}, p)}{\text{var}(\text{age})} \tag{5}$$

(Eq. 5) $m$: the slope of the linear regression line for the proportions of a given gene with age. See Eq. 3.

$$\Delta Max = \text{argmax}_{\text{age}}(p_{\max}) - \text{age}_{\max}$$

$$dMax(\Delta Max) = \begin{cases} 1 & \text{if } \Delta Max < 50 \\ -(\Delta Max - 50) & \text{otherwise} \end{cases}$$

$$aMax(\text{age}_{\max}) = \begin{cases} 1 & \text{if } \text{age}_{\max} \geq 38 \\ \text{age}_{\max} - 38 & \text{otherwise} \end{cases}$$

$$GAD = old(x) + gain(x) + young(x) + dMax(\Delta Max) \\ + aMax(\text{age}_{\max}) + m \times 5 \tag{6}$$

(Eq. 6) *GAD* : An individual human gene age-dynamic score is a sum of the weighted metrics. $\Delta Max$ is the difference in years between the population with the maximum positive gene proportion and the oldest age bin. Variables $old(x), gain(x), young(x)$ are from Eq. 3. Slope ($m$) is represented in percentage points and given a weight multiplier of 5.

## Identifying novel senescence signatures from mice

Age-dynamic genes for each tissue cell type were found as described above. Count data were subset by these genes and further subset to only include cells from mice > 21 m. Subsets with fewer than 100 cells

were not tested further. The count matrixes were binarized to represent cells by gene positivity. Every pairwise combination of genes was tested for Pearson's correlation (Eq. 7). To test the statistical significance, each pairwise comparison was randomly permutated 500 times. Pairwise correlations were kept if they had a positive r value and if their r value was at least 0.05 higher than the respective q99 (99th percentile) r value from the random permutations (Eq. 8). The filtered correlations were used to construct networks with NetworkX (https://networkx.org/). The Louvain algorithm was used to group genes into clusters. Network clusters with fewer than 5 genes or genes with no correlations were removed. Genes loosely connected to clusters were removed if they had fewer than $\log(n_{cluster\,genes})$ (Where $n_{cluster\,genes}$ is the number of genes in a Louvain cluster) connections to other genes in the network. The cleaned clusters are hereby referred to as hubs, and the aggregated hubs for each cell type are cell-specific signatures.

$$r_{i,j} = \frac{\sum_{k=1}^{n}\left(x_{k,i} - \bar{x}_i\right)\left(x_{k,j} - \bar{x}_j\right)}{\sqrt{\sum_{k=1}^{n}\left(x_{k,i} - \bar{x}_i\right)^2}\sqrt{\sum_{k=1}^{n}\left(x_{k,j} - \bar{x}_j\right)^2}} \tag{7}$$

(Eq. 7) $r_{i,j}$: Pearson's correlation coefficient for dynamic genes $i$ and $j$. Where $x_{k,i}$ represents the binary expression value of gene $i$ in cell $k$; $x_{k,j}$ is the binary expression value of gene $j$ in cell $k$; and n is the total number of cells in the population.

$$t = q_{99}\left(r_{\text{perm}(i,j)}\right) + 0.05$$

$$r_{i,j} > t \text{ and } r_{i,j} > 0 \tag{8}$$

(Eq. 8) $t$: significance threshold. Where $r_{\text{perm}(i,j)}$ represents the distribution of correlation coefficients for 500 random permutations of gene $i$ and $k$. $q_{99}$ represents the 99th percentile value of this distribution. The inequality depicts one criteria of gene selection based on $t$.

## Identifying novel senescence signatures from humans

Age-dynamic genes for each dataset-tissue-cell-type were found as described earlier. Count data were then subset by these genes and further subset to only include cells from patients 48 years of age or older. Significant correlations, networks, hubs, and signatures were generated similarly to those from mice.

## Novel signature comparison

Each signature or hub has a set of genes and corresponding weights for how many connections a gene shares with other genes. Pairwise cosine similarity was calculated by comparing the union of each gene list and imputing 0 s (Eq. 9). For pairwise hypergeometric similarity between two signatures, the cumulative distribution function for two lists of genes was determined using the genes present in the original species aggregated counts as the background list (Eq. 10). For signature network analysis, all pairwise hypergeometric sf values (i.e., *p*-values) were corrected with the Bonferroni method, converted to $-\log_{10}sf_{corrected}$, and used as similarity scores between signatures if they were significant.

To find genes represented in the signatures more than expected by chance, we used a random permutation method. A set of hubs with random genes identical in size to the original signatures were generated 1000 times from the background set of expressed genes in the dataset. A distribution was created representing the number of times each gene was found in each of the 1000 permutations. The actual number of signatures a gene was found in was compared to this distribution to determine what proportion of randomly sampled genes

were below it in rank.

$$cosine\left(L_i, L_j\right) = \frac{\sum_{k=1}^{n} a_{i,k} a_{j,k}}{\sqrt{\sum_{k=1}^{n} a_{i,k}^2} \sqrt{\sum_{k=1}^{n} a_{j,k}^2}}, where\ a_{i,k} = \begin{cases} w_i k\ if\ g_k \in L_i \\ 0, otherwise \end{cases}$$

(9)

(Eq. 9) $cosine\left(L_i, L_j\right)$: cosine distance between two signatures $L_i$ and $L_j$. Where $a_{i,k}$ represent the weight of gene $k$ in gene list $i$. Where n represents the total number of genes in the union of signatures $L_i$ and $L_j$.

$$sf = 1 - P\left(\left|L_i \cap L_j\right| - 1, |N|, |L_i|, |L_j|\right)$$
$$= 1 - \sum_{k=0}^{x-1} \frac{\binom{L_j}{k}\binom{N-L_j}{L_i-k}}{\binom{N}{L_i}}$$

(10)

(Eq. 10) $sf$: survival function of the hypergeometric distribution $P(x, N, I, j)$. Where $L_i$ and $L_j$ represent two gene lists and $N$ represents the cardinality of the background gene list, which is comprised of all genes in detected in the respective dataset. $x$ is equal to the cardinality of the $L_i$ and $L_j$ intersection minus 1.

### Gene set enrichment – GO, KEGG, transcription factor binding

We used the Enrichr python API gseapy[47] for gene set enrichment against the GO and KEGG databases (refs). The background set of genes used came from all expressed genes from their respective datasets. Only FDR-corrected *p*-values below 0.05 were considered significant. A custom "senescence" gene set was added which was comprised of the union between all literature-based senescence markers collected for this study and senMayo[20].

For transcription factor binding analysis, the regions 1000 bp upstream and 500 bp downstream of the transcription start sites were extracted for each gene in a gene list. JASPAR 2020 core vertebrate non-redundant position frequency matrices were us as the input motifs[48]. The extracted regions were examined for relative motif enrichment using the MEME-suite simple enrichment analysis.

Only Benjamini-Hochberg-corrected p-values below 0.05 were considered significant.

### Scoring cells using *SenePy*

Gene signatures are comprised of genes and their respective number of edges in their network (termed importance value). We developed *SenePy*, a lightweight and fast scoring algorithm specific for our gene sets that borrows from Seurat's AddModuleScore() and Scanpy's tl.score_genes(). *SenePY* is built in Python and integrates well with scanpy and anndata. *SenePy* has four core functions: load_hubs(), translator(), score_hub(), and score_all_cells(). The load_hubs() function initializes the hub object which includes the hubs themselves along with additional metadata, such as each hub's enrichment for known senescence genes. Depending on the input data and its respective reference, the optional translator() function can be used to harmonize gene symbols based on known gene aliases. The score_hub() function takes one input hub and anndata and returns a list of scores for each cell. The score_all_cells() takes one input hub and anndata and stratifies the data based on input categories, for example, to score individual cell types separately to avoid confounding the score.

The scoring happens in multiple steps. First, the mean is calculated for each gene in the dataset across all cells (Eq. 11). All genes are ranked by their mean and split into n_bins (default: 25) expression bins (Eq. 12). Next, n_ctrl_size (default: 50) background genes are selected for each input signature gene from its corresponding expression bin (Eq. 13). The counts data are then optionally binarized (default: True)

to represent the binary senescence cellular state and the gene-cell positivity from which the underlying networks were derived. Next, the counts are optionally amplified (Default: True) by their corresponding importance value from the input signature (e.g., if [*Cdkn2a*, 2] is in the signature all *Cdkn2a* values would be multiplied by 2) (Eq. 14). Then the cell-by-signature-gene matrix is averaged across the cell axis and subtracted from the mean of the cell-by-background matrix also averaged across the cell axis (Eq. 15).

$$m_j = (1/n)\sum_{i=1}^{n} X_{ij}$$

(11)

(Eq. 11) Where $X$ is a matrix which contains the expression level of gene $j$ in cell $i$ and $m_j$ is the average expression of gene $j$ across all cells.

$$B_1, B_2, \ldots, B_{n\_bins}$$

(12)

(Eq. 12) Where $B_{n\_bins}$ is the number of bins used to categorize every gene based on their mean expression.

$$s \in B_k, for\ s \in S$$

$$BG_{s^k} = \left\{g \mid g \in B_k^{(n_{ctrl\ size})}, g \neq s\right\}$$

$$BG = \bigcup_{s \in S} BG_{s^k}$$

(13)

(Eq. 13) $BG$: background gene set. Where $S$ is the gene signature and $s$ is a gene within $S$. $B_k$ is a subset of genes that fall within the k-th expression bin based on their mean expression. Where $g$ is a background gene selected from the expression bin $B_k$ and $n_{ctrl\ size}$ is the number of background genes selected for each signature gene $s$ from the corresponding expression bin. Where $BG_{s^k}$ is a set of background genes randomly selected from the same expression bin $B_k$ as the signature gene $s$. $BG$ is the union of all background genes selected for each signature gene $s$. Note, $BG$ depends on $B_k$ and set $S$.

$$Y_{ij} = \left\{1\ if\ X_{ij} > 0,\ 0\ otherwise\right\}$$

$$Z_{ij} = Y_{ij} * I_j$$

(14)

(Eq. 14) $Z_{ij}$: modified expression matrix. Where $Y_{ij}$ is the optionally binarized expression matrix $X_{ij}$ and $I_j$ represents the optional importance values for gene $j$. The optional importance values are obtained by the centrality of a gene in its network signature (i.e., the number of connected edges).

$$\overline{Y_{i,S}} = \frac{1}{|S|}\sum_{s \in S} Z_{i,s}, for\ i \in all\ cells$$

$$\overline{Y_{i,BG}} = \frac{1}{|BG|}\sum_{g \in BG} Z_{i,g}, for\ i \in all\ cells$$

$$Score_i = \overline{Y_{i,S}} - \overline{Y_{i,BG}}, for\ i \in all\ cells$$

(15)

(Eq. 15) $Score_i$: *SenePy* score for cell $i$. Where $|S|$ and $|BG|$ are the cardinality (number of elements in the set) of gene signatures $S$ and $BG$. Where $Z_{i,s}$ and $Z_{i,g}$ represent the optionally amplified expression values of the genes in the gene signature $S$ and background gene set $BG$, respectively.

## Merging multiple signatures and identifying a universal senescence signature

The *senePy* signature database is comprised of multiple gene sets of different sizes. An individual gene may be found in multiple sets. The universal signature can be defined by finding genes that are over-represented in the signature gene sets using all genes in the respective species dataset as the background set. We can determine the probability a gene is included in any number of sets using a generating function (Eq. 16). The cumulative distribution function is calculated from the cumulative sum of the resulting probabilities. Finally, the *p*-value for any given number of sets by subtracting the respective cdf value from 1. The resulting *p*-values are then corrected using the Benjamani-Hochberg

$$p_i = \frac{|S_i|}{|BG|}$$

$$G(x) = \prod_{i=1}^{k} \left[ (1 - p_i) + p_i x \right]$$

$$G(x) = \sum_{m=0}^{k} a_m x^m$$

$$P(x = m) = \frac{a_m}{\sum_{j=0}^{k} a_j} \tag{16}$$

(Eq. 16) $P(x = m)$: The probability that a gene is found in exactly $m$ sets (gene signatures). $p_i$ is the probability a gene is found in each set $S_i$ based on the number of genes in the background $BG$. $G(x)$ is the overall generating function of all $k$ sets and can be defined as the product of individual generating functions for each set. $G(x)$ is expanded into a polynomial where $a_m$ is the coefficient of $x^m$ and represents the probability a gene is in $m$ sets. The probability mass function is obtained by normalizing the probabilities so the sum of probabilities is 1.

## Senescence burden in spatially resolved transcriptomics

Data was preprocessed in Scanpy and spots with fewer than 1000 detected genes were removed. Cells were normalized to 10,000 counts and log converted. The 8 heart-specific mouse hub signatures were used to score the spatially resolved mouse hearts independently using senepy.score_hub() with a translator() and with binarize and importance set to False because Visium data has higher gene counts than single-cell data. Outlier spots were identified in each sample if they fell 3 standard deviations outside the mean for their respective sample distribution in addition to a combined sample distribution ($Outlier > \mu + 3\sigma$). The outliers from each signature were merged to determine if any given spot was an outlier. Relative senescence burden is presented as the proportion of outlier spots. For the mouse brains, we used the top 150 most common genes in all the signatures because we had no specific mouse brain signatures. To determine spatial autocorrelation, we used the ESDA Python package (https://pysal.org/esda/). The weights of the autocorrelation were weighted by the inverse of the Euclidean distance between two spots with a value of 1 to denote an outlier and 0 for normal spots. Three is used as a maximum value for Euclidean distance and the weights for distances beyond three are set to 0 (Eq. 17).

$$I = \frac{n \sum_i \sum_j \left( \frac{1}{d(i,j)} \right) \cdot \delta(d(i,j) \le 3)(x_i - \bar{x})(x_j - \bar{x})}{\sum_i \sum_j \left( \frac{1}{d(i,j)} \right) \cdot \delta(d(i,j) \le 3) \sum_i (x_i - \bar{x})^2}, \text{p} - \text{value} = 1 - (\Phi(I))$$

$$\tag{17}$$

(Eq. 17) $I$: Moran's I. Where $n$ is the number of spots; $d(i,j)$ is the Euclidean distance between spot $i$ and spot $j$; $\delta(d(i,j) \le 3)$ is 1 if the distance is greater or equal to 3 and otherwise 0; and $x_i$ and $x_j$ are the values at spot $i$ and spot $j$. $\Phi(I)$ is the CDF of the standard normal distribution at the Moran's I value.

## Senescence burden in COVID-19 mortality

Single-cell lung data from 20 patients that died from COVID-19 and 7 control patients were collected from an available atlas[35]. Doublets were removed from each individual sample using SOLO[49] in combination with SCVI-tools. Cells with low counts or high mitochondrial reads were removed. SCVI tools were used to integrate the 27 samples, using sample ID as a categorical covariate and mitochondrial read percent, ribosomal read percent, and total counts as continuous covariates. Cell types were manually annotated using known cell-type markers (PanglaoDB). Cell types were scored with respective cell type hubs from *SenePy* (e.g., epithelial cells were scored with ciliated epithelial, basal cell, club cell, and pneumocyte hubs) using the senepy.score_all_cells() function. Cells were divided and scored as individual subtypes (e.g., AT1, AT2, airway epithelium). Cell outliers were identified in each sample if they fell 3 standard deviations outside the mean within every respective cell-type distribution (($Outlier > \mu + 3\sigma$). Outliers were merged across hubs to identify all cells with potential senescence burden and output as a proportion of total cells.

## Validation of senePy signatures from bulk-RNA data

Data were obtained in mixed formats due to differences in study data availability. If differential expression data were available, they were used directly in gene set enrichment. Otherwise, raw counts were processed through a standard Deseq2[50] pipeline implemented with pyDeseq2[51]. Enrichment analyses were done with the gseapy UCSD GSEA API[52]. Only senePy signatures relevant to the respective context were used in each enrichment comparison. For example, only *SenePy* lung signatures were tested if the data came from the lungs. The senGPT signature was generated by prompting ChatGPT-4 for 100 upregulated gene markers of cellular senescence in two non-overlapping batches of 50 genes.

## Pseudotime analysis

Raw cell and hashtag counts were retrieved for GSE222338[33]. The data were demultiplexed using HashSolo[49]. Cells were filtered if they were outside of 5 median absolute deviations from their respective log1p_total_counts, log1p_n_genes_by_count, pct_count_in_top_20_genes, and pct_counts_mt distributions. Doublets were removed with scrublet[53]. Psuedotime was calculated using CellOracle with a centrally located (2D space) root cell in the mVenus control group[54].

## Reporting summary

Further information on research design is available in the Nature Portfolio Reporting Summary linked to this article.

# Data availability

We used multiple publicly available datasets to develop, validate, and apply *SenePy*. Under the Gene Expression Omnibus, they can be found at GSE132042 (*tabula muris senis*), GSE201333 (*tabula sapiens*), GSE115469 (human liver), GSE134355 (human cell landscape), GSE185553 (human hippocampus), GSE185277 (human hippocampus), GSE198323 (human hippocampus), GSE121611 (human lung), GSE122960 (human lung), GSE155182 (p16 reporter), GSE222951 (hepatocyte tumorigenesis), GSE130727 (in vitro senescence), GSE180750 (lung senolysis), GSE184348 (muscle senolysis), GSE176092 (heart spatial), GSE148612 (brain spatial), and GSE171524 (lethal COVID-19). Human skin aging data is available in the Genome Sequence Archive under HRA000395. Human heart data are available at www.heartcellatlas.org. The data generated in this study are

provided in the Supplementary Information and the Source Data file available at https://doi.org/10.5281/zenodo.14775758.

## Code availability

All the code used in the analysis of this manuscript and the *SenePy* source code can be found at https://github.com/jaleesr/senepy. The original release for this publication can be found at https://doi.org/10.5281/zenodo.14775736.

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

## Acknowledgements

The studies were supported by NIH grants R01AG091545 (J.R.), R01-HL163978(J.R.), P01HL160469 (to J.R.) R01-HL152515 (to J.R.), T32-HL139439 (to M.A.S.), F31-AG090005 (to M.A.S). BioRender was used to make Figs. 1a, 2a, and Supplementary Fig. 2a. BioRender icons were used in Figs. 1d, 1f, 2b, 2d, 4e, 5b, 5d, 6b, 6d, and 6e as well as Supplementary Fig. 6a–e All icons can be found at Sanborn, M. (2025) https://BioRender.com/l04t362.

## Author contributions

M.A.S and J.R. conceived the project, designed the *SenePy* platform, and supervised the research. M.A.S. performed the analyses and wrote the original draft. X.W., S.G., and Y.D. provided input regarding the algorithm development and implementation of *senePy*. All authors reviewed the manuscript and made revisions.

## Competing interests

The authors declare no competing interests.
