## [Transparent Peer Review file · Nature Communications]

Unveiling the Cell-Type Specific Landscape of Cellular Senescence through Single-Cell Transcriptomics using SenePy

Corresponding Author: Dr Jalees Rehman

Version 0:

Reviewer comments:

Reviewer #1

(Remarks to the Author)

Sanborn et al. described an analytical framework for identifying age-associated genes from large-scale public single-cell datasets spanning different ages in mice and humans. Using this analytic strategy, the authors identified de novo cell-type-specific senescent marker genes that had not been revealed by previous experimental studies. They further demonstrated the biological relevance of these signatures by clustering them into sub-gene hubs and showing differential expression patterns of these hubs during aging. Finally, the authors validated that their gene sets better describe senescence compared to existing senescent markers through in vitro senescence induction experimental datasets, senescence marker-enriched mouse single-cell data, and spatial transcriptomes from mouse colon and brain.

The topic of identifying senescence-associated gene signatures is of significant importance for aging research, and I appreciate the goal of achieving this by mining large-scale single-cell datasets with rich metadata. Following is a list of comments that should be addressed before its publication.

Major Points:

- 1. Analytical Strategy:** The approach for identifying senescent markers lacks quality control and benchmarking. The authors employ deterministic cutoffs, such as considering genes expressed in less than 5% of young cells and between 0.5% and 20% of old cells as candidate signature genes. Additionally, the criteria for gene expression increase, such as a fold change greater than 2.5 or a positive proportion gain larger than 5%, need quantitative validation and clearer visualization to demonstrate their feasibility. Furthermore, the biological rationale behind these specific patterns of senescent marker expression requires clarification. The authors should consider using weight-based soft thresholds instead of the current hard cutoffs.
- 2. Statistical Significance:** The manuscript lacks statistical rigor in several sections, undermining the credibility of the results. For instance, the first section critiques differential gene expression for senescent marker identification but then uses metrics without considering statistical significance. The gene filtering step described in point 1 is particularly problematic as it defines aging-associated genes based solely on numerical increase. Similarly, figures like Fig 1C and Fig 6I-L show CDKN2A+ cell proportion increases without replicates or significance tests.
- 3. Discussion on Differential Expression:** The discussion in the first section about the limitations of differential expression is unexpected. Given the sparsity of single-cell data, gene expression often appears binarized (0/1). The authors' results from cell positivity may not be distinct from differential expression, potentially sacrificing statistical stringency for sensitivity. A thorough evaluation and benchmarking between these two strategies are essential before proceeding.
- 4. Gene Hub Identification:** The identification of gene hubs is preliminary and lacks biological interpretation. The choice of NetworkX for network construction needs justification. Have other network construction methods been benchmarked? Have multi-omics or epigenetic data been integrated for better interpretation? A concrete example showing that hub genes are enriched in a gene regulatory network or share regulatory mechanisms is necessary. Relying solely on GO term analysis for functional annotation is insufficient.

5. Choice of Cell Types for Comparative Analysis: The choice of tongue keratinocytes as a basis for comparative hub-0/1 analyses across cell types is questionable. Using a specific cell type to check enrichment in others may be uninformative. A universal hub-0/1 signature should be identified, with its cross-cell type sharing examined.

6. Manuscript Organization: The manuscript's organization needs improvement. There are redundant interpretations in sections that would fit better elsewhere. For example, starting from line 192, the examination of gene signature similarities across cell types is misplaced under the sub-title "Distinct modes and phenotypes of senescence exist within the same cell types". Additionally, the "SenePy signatures" mentioned in Fig 2E reappear only at line 181. Discussions on cell-type-specific senescent kinetics (Fig 5) overlap with similar concepts in Fig 3H. The term "senescence burden" (line 341) lacks explanation.

Minor Points:

1. The manuscript does not clearly distinguish between aging and senescence. Aging refers to macroscopic organismal changes over time, while senescence describes cellular function decline. More discussion and clarification are needed, especially since the authors use aging single-cell datasets to study senescence.
2. Proliferating cells are identified using Mki67 positivity. However, cell cycle scoring based on gene programs is a more common approach given the sparsity of single-cell data.
3. In the validation section (Fig 6), the enrichment of differentially expressed genes in the authors' gene features versus other published senescence markers needs clarification. Were these top features manually selected based on the cell type of corresponding datasets, or was an unsupervised enrichment analysis performed?
4. The colocalization between Col1a1 expression level and senescence burden mentioned in line 360 requires clarification. If Col1a1 is part of the heart senescence signature, this colocalization might be trivial.
5. Instead of highlighting senescent outlier spots in Fig 7, plotting the senescence score of each spot over the histology image might better reflect the overall trend of cellular senescence. Statistical outliers may not necessarily indicate biological significance.
6. The visualization in the manuscript needs improvement. For example, the heatmaps in Fig 1 are abstract and unclear. More suitable visualization methods should be considered for Fig 2C-D. Including cell types common to different tissue types is reasonable, but expecting brain cells in the bladder is not.

(Remarks on code availability)

The GitHub website lacks key tutorials and instructions about how to use the package in real applications. This should be improved.

Reviewer #2

(Remarks to the Author)

(Remarks on code availability)

Reviewer #3

(Remarks to the Author)

The paper analyzed multiple scRNA-seq aging datasets to identify cell-specific signatures of cellular senescence, resulting in 75 mouse and 65 human senescence signatures. The authors further explored the temporal and transcriptional characteristics of these signatures. They developed SenePy, a tool that utilizes these signatures to score cells for cellular senescence, outperforming comparable methods. Additionally, the authors used SenePy to quantify cell senescence and investigated its relevance to various diseases.

The paper provides a thorough study of markers for cellular senescence. The identified markers and the cell-scoring package will be beneficial to the single-cell research community. However, the data analyses and statistical rigor could be further enhanced. My specific comments are as follows.

- Line 107: what's the hypergeometric p-value for? The intersection symbol before "=52" is too informal. Can you revise it? Also, $125+110-52$ is not equal to 181, so I am not sure about these numbers.

- Line 109: what does "cell-specific" mean? Does it mean "cell type-specific"?

- Line 110: how were the 76 cell-specific signatures defined? Zhang et al. inferred cell type-specific aging markers for different cell types separately. I don't think there is a set of 76 cell-specific signatures.
- Line 119: it is not clear why the small overlap between the aging markers in Zhang et al. eLife 2021. (line 111) and the 181 curated experimentally validated CS markers (lines 106-107) suggest DGE analysis is not suitable for identifying universal aging markers. First, are the 181 curated experimentally validated CS markers expected to be shared across cell types? Second, there are alternative methods using DGE analysis to identify aging markers.
- Line 123: the term "cell positivity" needs to be clarified.
- Line 136: the lack of overlap may not indicate the heterogeneity of CS markers. It may be just due to the high noise level of scRNA-seq data.
- Line 149: it is not clear if the proposed method for defining cell type-specific signatures is better than the method in Zhang et al. eLife 2021.
- Line 152-513: is there a reference for this prior knowledge: "senescent cells accumulate with increasing organismal age but remain a minority population"
- Line 149: the authors should provide more details and justification for the proposed method for defining cell type-specific signatures. The procedures appear ad-hoc to me. How are the networks constructed? How are the CS signatures selected from the network of hubs? Why is this method preferred over DGE, which is more conventional? Systematic benchmarking experiments comparing the proposed method with existing methods are needed.
- Line 154: what's a "cell-by-gene positivity matrix"? I think it refers to the binarized count matrix but it should be clarified here. Also, the authors need to provide justifications for binarization. Why is the better than the original count matrix?
- Line 161: how can we be sure that this negative result is not just because of noise? The lack of overlap seems to be a negative result, and, in general, statistical analysis cannot draw conclusions about negative results (in other words, $P < 0.05$ indicates significant effects but $P > 0.05$ doesn't indicate no effects).
- Line 162-164: this may be due to the batch effect between tissues. Have the authors verified that this is not the case?
- Line 165-166: can you quantify how prevalent this observation is? Anecdotal examples are helpful but a rigorous quantification is necessary.
- Line 176: the authors concluded that "the previously defined set of unspecific senescence markers was more closely associated with the newly derived signatures". However, it is not clear what the statistical analysis is and what's the strength of evidence, such as a p-value.
- Line 181-183: it is not clear why SenePy is novel. How it is different from applying the Seurat cell-scoring module to the set of signatures identified by this paper? Also, there are alternative cell-scoring methods with better calibration, such as scDRS (Zhang et al. Nat Genet 2022).
- Line 574-578: did the authors consider confounding when computing the gene-gene correlation?
- Line 192-193: Please provide p-values for the finding "For example, aging mouse tongue keratinocytes consisted of two hubs, both enriched in established CS marker genes". In general, all findings need to be supported by statistical evidence.
- Line 192: why did you choose mouse tongue keratinocytes? How representative is this example? Anecdotal examples are helpful but a systematic quantification of the overall patterns across cell types will provide deeper insights.
- Line 206: how was the outlier threshold determined? Would FDR be more appropriate? I note that alternative cell-scoring methods such as scDRS (Zhang et al. Nat Genet 2022) can produce cell-level p-values.
- Line 207: can you provide a p-value for "increased significantly"?
- Line 212: how was fibroblasts chosen as an example? How representative is this example? Anecdotal examples are helpful but a systematic quantification of the overall patterns across cell types will provide deeper insights.
- Line 230: the identification of universal senescence signatures may be confounded by the composition of cell types in your datasets.
- Line 230: how do the universal senescence signatures identified here compare to the 330 global aging genes in Zhang et al. eLife 2021?
- Line 234: what is n in " $n \geq 12$ "? How was the threshold 25% and the p-value threshold 0.0014 determined?

- Line 241: how was the threshold of 8 chosen?
- Line 251: how consistent are the universal senescence signatures between human and mouse data?
- Line 256: multiple testing correction is needed instead of a p-value threshold of 0.05.
- Line 283-284: Have the authors considered confounding for explaining the between-patient difference? The lack of overlap seems to be a negative result, and, in general, statistical analysis cannot draw conclusions for negative results (in other words, $P < 0.05$ indicates significant effects but $P > 0.05$ doesn't indicate no effects).
- The paper created many different types of gene signatures and it is very easy to get lost. I suggest creating a table containing all signature types, such as the 181 experimentally validated CS marker genes (line 107), aging signatures from Zhang et al. eLife 2021, and the cell type-specific signatures identified by this paper.

Minor:

Line 107: "maker"  "marker"

- Line 154: "matrixes"  "matrices"

(Remarks on code availability)

Reviewer #4

(Remarks to the Author)

Abstract

The final paragraph is too technical and should be rewritten for clarity for the broad audience of Nature Communications.

Introduction

Heterogeneity is difficult to only characterize via CDKN2A. Reporter mice for CDKN1A, for instance, are also gaining popularity in the senescence field, and the role of Cdkn2a is questioned in specific conditions and locations. Perhaps there are more markers than only these two.

Results

Not finding a universal senescence marker gene set does not automatically show that there are none for specific cell types that may or may not be tissue-specific.

What is the biological rationale for similar gene expression signatures more likely to be found in cells from the same tissue vs. cell type?

The increase of the SenPy senescence score with aging in Fig. 3E is quite convincing.

Temporal kinetics can be assessed via RNA velocity, as shown by Manno et al. (Nature 2018). This would be a perfect (and the only unbiased) way to demonstrate the superiority of your senescence signature within scRNA-Seq signatures.

The demonstration of each pathway, like in Fig. 4B, does not substantially contribute to the overall story.

line 292-295 and 302-303: Without experimental evidence, this statement is far beyond the verification provided.

The p16+ cells association is also a huge limitation. Maybe SenePy just identifies p16-associated senescence reliably?

The spatial datasets do not add substantially to the overall story. In particular, why is COVID-19 important? The association with cellular senescence is well-known.

The kinetics of cellular senescence has not been elucidated. Particularly, velocity should be used in the tabula muris senis dataset to verify that SenePy can identify a progress of cellular senescence along time (and age) in a specific cell type (tissue).

Discussion

The discussion needs to be shortened substantially. While in the upcoming era of senolytics, SenePy may be useful, you failed to prove that it is also worth considering in a senolytic population.

Overall, experimental validation of SenePy (i.e. with in vitro cellular senescence over a long time) would help the manuscript.

Also, a senolytic cohort that shows a reduction of SenePy would be highly appreciated.

(Remarks on code availability)

The overall code looks correct.

However, the README file did not contain enough instructions for installing and running the application.

Subsequently, I was not able to install and run the code.

Version 1:

Reviewer comments:

Reviewer #1

(Remarks to the Author)

The authors have addressed all the concerns raised effectively and have made appropriate amendments to the manuscript. I am satisfied with the revisions and believe that the manuscript is now suitable for publication.

(Remarks on code availability)

The code provide a readme file with enough instructions in installing and running

Reviewer #2

(Remarks to the Author)

(Remarks on code availability)

The github tutorial is well organized.

Reviewer #3

(Remarks to the Author)

Thank you for the detailed and thorough response, as well as the additional experiments. Most of my concerns have been addressed. Below are my follow-up questions.

- Line 165-166: can you quantify how prevalent this observation is? Anecdotal examples are helpful but a rigorous quantification is necessary.

- Rv1: Still, I think a quantitative assessment is warranted. Would something like "average overlap between the same cell type from different tissues" and "average overlap between different cell types from the same tissue" work?

- Line 283-284: Have the authors considered confounding for explaining the between-patient difference? The lack of overlap seems to be a negative result, and, in general, statistical analysis cannot draw conclusions for negative results (in other words, $P < 0.05$ indicates significant effects but $P > 0.05$ doesn't indicate no effects).

- Rv1: Still, the pattern might be driven by patient-specific factors other than age, such as sex or dietary habits. If investigating this is challenging, it warrants further discussion.

(Remarks on code availability)

Reviewer #4

(Remarks to the Author)

All my concerns have been addressed.

(Remarks on code availability)

The code for every figure available and can be run relatively easy.

Responses to Reviewers:

Reviewer #1 (Remarks to the Author):

Sanborn et al. described an analytical framework for identifying age-associated genes from large-scale public single-cell datasets spanning different ages in mice and humans. Using this analytic strategy, the authors identified de novo cell-type-specific senescent marker genes that had not been revealed by previous experimental studies. They further demonstrated the biological relevance of these signatures by clustering them into sub-gene hubs and showing differential expression patterns of these hubs during aging. Finally, the authors validated that their gene sets better describe senescence compared to existing senescent markers through in vitro senescence induction experimental datasets, senescence marker-enriched mouse single-cell data, and spatial transcriptomes from mouse colon and brain.

The topic of identifying senescence-associated gene signatures is of significant importance for aging research, and I appreciate the goal of achieving this by mining large-scale single-cell datasets with rich metadata. Following is a list of comments that should be addressed before its publication.

Major Points:

1. Analytical Strategy: The approach for identifying senescent markers lacks quality control and benchmarking. The authors employ deterministic cutoffs, such as considering genes expressed in less than 5% of young cells and between 0.5% and 20% of old cells as candidate signature genes. Additionally, the criteria for gene expression increase, such as a fold change greater than 2.5 or a positive proportion gain larger than 5%, need quantitative validation and clearer visualization to demonstrate their feasibility. Furthermore, the biological rationale behind these specific patterns of senescent marker expression requires clarification. The authors should consider using weight-based soft thresholds instead of the current hard cutoffs.

We thank the reviewer for this comment. We have removed hard thresholds and have instead switched to a weighted approach. Now, each individual gene has a score that is the sum of multiple weighted values. Instead of hard thresholds, an individual value is a function of how close individual metrics are to an ideal range. These ranges, like the now deprecated hard thresholds, are based on prior biological knowledge: 1) We expect senescent cells to increase in proportion with organismal age and 2) to be a minor fraction even in aged organisms. These observations can be extrapolated into metrics for each gene: starting proportion, ending proportion, old-to-young proportion gain, and the old-to-young proportion ratio (schematic step #2).

Starting: We set an ideal range of 0-5% of cells expressing a gene in young organisms to select genes that are specific to cellular senescent and not expressed at appreciable rates during baseline conditions or in young organisms. The starting score decreases after 5%.

Ending: Idda et al. 2020 have stained for senescence markers in multiple tissues at multiple ages and found that most are expressed at rates below 20%. We have set an ideal ending range of 20% or below in aged organisms. The ending score decreases at higher than 20%.

Gain: Senescent cells increase in proportion with age. We score this metric based on the increase in the proportion of a gene between the young and old population. The metric score is

lower than 1 if the gain is low or higher than what we expect (i.e., the expected ending proportion should be lower than 20%).

Ratio: Senescent cells increase in proportion with age. Higher proportional ratios would be expected in genes specific to cellular senescence. The ratio score is linear to the observed old-young ratio but capped to reduce the impact on genes imputed at young ages. The ratio is indispensable for senescence programs and genes expressed at low rates and have lower relative gain scores. For example, *Cdkn2a* often has a high ratio but a relatively low gain score.

Previously, a gene was removed if any of the metrics were above/below a set hard threshold. Now, a gene can be outside an ideal range but still considered significant if the other metrics contribute strongly to the overall score. For example, a gene may still have a significantly high score if it is found in 21% of old cells if the other metrics are also close to their ideal range (it would have been removed previously due to a hard threshold of 20%). The combined score is compared to a null distribution to determine how significant the observation is. The functions to determine the score and this new adaptive approach are described in detail in the updated Methods section and outlined in the updated schematic.

Importantly, the majority of genes identified by the previously fixed thresholds were again identified with this new adaptive threshold approach. The resulting gene output is stable even with moderate changes to the underlying weighted functions (e.g., slopes, ideal range, etc.). A small number of new genes were included or removed but have not affected downstream analysis or *senePy* scoring significantly. No results that were dependent on *senePy* signatures changed substantially. All major findings and validations of the previous version remain intact. An additional single-cell validation based on *in vivo* senescent hepatocytes included in the revision also strongly supports the generated signatures. These data support the overall robustness of the pipeline.

2. Statistical Significance: The manuscript lacks statistical rigor in several sections, undermining the credibility of the results. For instance, the first section critiques differential gene expression for senescent marker identification but then uses metrics without considering statistical

significance. The gene filtering step described in point 1 is particularly problematic as it defines aging-associated genes based solely on numerical increase. Similarly, figures like Fig 1c and Fig 6i-l show CDKN2A+ cell proportion increases without replicates or significance tests.

By changing our method for determining what is a dynamic gene based on the reviewer suggestions (see above) we can now provide multiple testing corrected p values for each gene-cell type combination. We created a null distribution for each cell type by randomly shuffling the values within a cell-gene expression vector across the cells. This ensures that the null distribution accounts for confounders, such as gene sparsity. The null distribution for each cell type was created from around 20 million random permutations. The p-value is determined by comparing observed gene-cell scores to the null distribution. The genes identified in figures such as those in Fig. 1B/E/F were not significantly impacted because the thresholds we used before were strong enough to account for random chance. We have updated the figures and included corrected p values. The figures below show modified versions from the first submission now with FDR-corrected p values. In E/F, only significant genes are now shown.

Fig 1B

Fig 1C/D pt. 2, S1A pt. 2

Fig 1E

Fig 1F

3. Discussion on Differential Expression: The discussion in the first section about the limitations of differential expression is unexpected. Given the sparsity of single-cell data, gene expression often appears binarized (0/1). The authors' results from cell positivity may not be distinct from differential expression, potentially sacrificing statistical stringency for sensitivity. A thorough evaluation and benchmarking between these two strategies are essential before proceeding.

We previously looked at differential expression in these same data examined their ability to identify senescence markers. Zhang et al. performed differential expression on these data and found robust organismal aging signatures. However, these signatures contained a paucity of known senescence markers. We show these findings in **Figure 1a, 1b, S2a, S2b** and at several points in the first section of the results.

We have added an additional analyses and figures to **Supplemental Figure S2** in the revision (**Fig S2D,E**) that directly compares the enrichment of known senescence markers in the genes we derived versus the aging signatures derived from differential expression. 20% of the DE-derived aging signatures are enriched for known senescence markers compared to 32% for our cellular senescence signatures. We also compare a universal organismal aging signature derived by differential expression (Global aging genes: GAG) to our *senePy* universal signature (**Fig S2E**). Strikingly, the *senePy* universal signature is highly enriched for known cellular senescence markers ($p = 1.4 \times 10^{-13}$) and the DE-derived signature is not. Interestingly, there is less overlap between the DE GAG set and the universal *senePY* signature than what is expected (**Fig S2F**), indicating that these methods are uncovering biologically distinct phenomena (i.e., organismal aging vs. cellular senescence).

Furthermore, not a single DE-aging signature identified by Zhang et al. contained *Cdkn2a* even though this is considered one of the most universal markers of cellular senescence. In contrast, 20 of the 63 gene sets using our methodology contain *Cdkn2a*. Our goal is not to disparage differential expression, but to instead demonstrate that our method is better at finding genes specific to cellular senescence. Without a ground truth, we cannot further compare the two. Additionally, our revised method has stronger statistical backing and provides p values suggested by the reviewer.

4. Gene Hub Identification: The identification of gene hubs is preliminary and lacks biological interpretation. The choice of NetworkX for network construction needs justification. Have other

network construction methods been benchmarked? Have multi-omics or epigenetic data been integrated for better interpretation? A concrete example showing that hub genes are enriched in a gene regulatory network or share regulatory mechanisms is necessary. Relying solely on GO term analysis for functional annotation is insufficient.

The biological interpretation of gene hubs is that they are groups of genes that are more likely to be co-expressed; i.e., more likely to be expressed in the same cells. Co-expression is a commonly examined phenomenon in transcriptomics and has well-known biological significance. NetworkX is only used for visualization and to easily parse the network data structure. In this scenario, the tool used for network construction only impacts visualization and can be accomplished independently of NetworkX or any other tool. At the core of our network analyses, we use well-known and standard methods, such as phi coefficient analysis and Louvain clustering. Integrating additional multiomics data is a great idea, but that would require multiomics single-cell data with the similar biological context; e.g., a ground truth *in vivo* senescence dataset, a multiomics dataset within the same tissues/cells at the same ages which currently are not available for the robust range of ages that we examined. We do examine regulatory elements of each signature based on transcription factor binding enrichment (**Fig 4**). We also validate these hubs against several different datasets and have included an additional validation in the revision (**Fig 6 F-L**).

5. Choice of Cell Types for Comparative Analysis: The choice of tongue keratinocytes as a basis for comparative hub-0/1 analyses across cell types is questionable. Using a specific cell type to check enrichment in others may be uninformative. A universal hub-0/1 signature should be identified, with its cross-cell type sharing examined.

The keratinocytes were chosen for two primary reasons: 1) there is more than one distinct hub in the signature and 2) there is a large number of keratinocytes across the tested ages, which helps to account for age-to-age noise. Additionally, both hubs were highly enriched for known senescence markers. Other cell types had similar temporal trends and network topology, but it is not representative of all hubs. This has been clarified in the text and we have added additional descriptive analysis of every mouse hub into the supplement and text of the manuscript. New figures **S4A ,C** show the network structure of every signature and the proportion of cells expressing these signatures in young and old mice in the respective cell type. Figure **3A** shows that there is a general increase in the cells expressing these signatures with age.

Fig S4A

Fig S4C

Fig 3A

6. Manuscript Organization: The manuscript's organization needs improvement. There are redundant interpretations in sections that would fit better elsewhere. For example, starting from line 192, the examination of gene signature similarities across cell types is misplaced under the sub-title "Distinct modes and phenotypes of senescence exist within the same cell types". Additionally, the "SenePy signatures" mentioned in Fig 2E reappear only at line 181. Discussions on cell-type-specific senescent kinetics (Fig 5) overlap with similar concepts in Fig 3H. The term "senescence burden" (line 341) lacks explanation.

We have made changes throughout the manuscript to improve the clarity.

Minor Points:

1. The manuscript does not clearly distinguish between aging and senescence. Aging refers to macroscopic organismal changes over time, while senescence describes cellular function decline. More discussion and clarification are needed, especially since the authors use aging single-cell datasets to study senescence.

We agree that aging and cellular senescence are distinct phenomena. We stressed this at multiple points in the manuscript, but we also agree that we could be more careful when discussing this (e.g., referring to it as cellular senescence instead of just senescence). We have revised the text accordingly for the sake of consistency.

2. Proliferating cells are identified using Mki67 positivity. However, cell cycle scoring based on gene programs is a more common approach given the sparsity of single-cell data.

Interestingly, we had originally performed both cell cycle scoring and Mki67 positivity. The proportion of Mki67 and the cell cycle score were very correlated ($p = 6.3 \times 10^{-17}$), and neither had a positive correlation to cellular senescence. We chose to show only MKi67 due to redundancy. In the revision, we have additionally included a regression plot (Fig S2C pt. 2) of the cell-cycle score versus the senescence signature score for each cell population and updated the text.

3. In the validation section (Fig 6), the enrichment of differentially expressed genes in the authors' gene features versus other published senescence markers needs clarification. Were these top features manually selected based on the cell type of corresponding datasets, or was an unsupervised enrichment analysis performed?

We only included analyses of gene signatures that were from the same or similar tissue context. We have included a section in the methods upon revision: "Validation of senePy signatures from bulk-RNA data".

4. The colocalization between Col1a1 expression level and senescence burden mentioned in line 360 requires clarification. If Col1a1 is part of the heart senescence signature, this colocalization might be trivial.

Col1a1 is not present in any of the heart signatures that were used to score the spots. Moreover, the expression of Col1a1 is not localized only to *senePy*-identified spots but is instead found in the surrounding spots as well. We have clarified this in the text.

5. Instead of highlighting senescent outlier spots in Fig 7, plotting the senescence score of each spot over the histology image might better reflect the overall trend of cellular senescence. Statistical outliers may not necessarily indicate biological significance.

We have analyzed both the binary outlier spots and the raw scores for each spot, but they tell a similar story (i.e., auto-correlation/clustering, colocalization with Col1a1). We chose to show the outliers because we think this better reflects a distinct cell state of cellular senescence as opposed to a stressed or pre-senescent state.

6. The visualization in the manuscript needs improvement. For example, the heatmaps in Fig 1 are abstract and unclear. More suitable visualization methods should be considered for Fig 2C-D. Including cell types common to different tissue types is reasonable, but expecting brain cells in the bladder is not.

The original dot plots (heatmaps) in **Fig 1** were objective analyses of the increase in senescence marker genes over time in different cell types/tissues. They did lack a statistical test, which we have remedied in this revision as suggested by the reviewer (addressed above). We hope that with the presence of a p-value, they are no longer considered abstract. **Fig 2c-d** do not compare cells between tissues, they compare *senePy* signatures. There are no brain cells in the bladder, but these are simply comparisons of cellular senescence signatures across distinct cell types

Reviewer #1 (Remarks on code availability):

The GitHub website lacks key tutorials and instructions about how to use the package in real applications. This should be improved.

We have substantially expanded the readme file and added to the tutorial notebook on Github. We have also added additional Github tutorials and included instructions on how to use *senePy* in R.

Reviewer #2 (Remarks to the Author):

Reviewer #3 (Remarks to the Author):

The paper analyzed multiple scRNA-seq aging datasets to identify cell-specific signatures of

cellular senescence, resulting in 75 mouse and 65 human senescence signatures. The authors further explored the temporal and transcriptional characteristics of these signatures. They developed SenePy, a tool that utilizes these signatures to score cells for cellular senescence, outperforming comparable methods. Additionally, the authors used SenePy to quantify cell senescence and investigated its relevance to various diseases.

The paper provides a thorough study of markers for cellular senescence. The identified markers and the cell-scoring package will be beneficial to the single-cell research community. However, the data analyses and statistical rigor could be further enhanced. My specific comments are as follows.

- Line 107: what's the hypergeometric p-value for? The intersection symbol before "=52" is too informal. Can you revise it? Also, $125+110-52$ is not equal to 181, so I am not sure about these numbers.

The hypergeometric P-value is to show that there is significant overlap between the SenMayo gene set and an independently generated gene set from literature review.

We appreciate the reviewer for catching this discrepancy, which arose because the human literature marker set has two fewer genes than the mouse set. We have expanded and corrected this section.

- Line 109: what does "cell-specific" mean? Does it mean "cell type-specific"?

Revised to "cell-type-specific".

- Line 110: how were the 76 cell-specific signatures defined? Zhang et al. inferred cell type-specific aging markers for different cell types separately. I don't think there is a set of 76 cell-specific signatures.

We agree that Zhang et al. defined signatures for 76 separate cell types. I think the confusion here arose from the wording. We have revised "76 cell-specific signatures" to "76 cell-type-specific signatures" for clarity.

- Line 119: it is not clear why the small overlap between the aging markers in Zhang et al. eLife 2021. (line 111) and the 181 curated experimentally validated CS markers (lines 106-107) suggest DGE analysis is not suitable for identifying universal aging markers. First, are the 181 curated experimentally validated CS markers expected to be shared across cell types? Second, there are alternative methods using DGE analysis to identify aging markers.

We agree that DGE analysis between old and young cell populations, as performed by Zhang et al., is a suitable way to identify organismal aging markers. However, there is an important distinction here between organismal aging and cellular senescence. In this section, we point out that the DGE analysis performed using the same data does not identify cellular-senescence-associated genes that are clearly dynamic in these populations. We believe this is because cellular senescence affects a small number of individual cells as opposed to the population overall and because cellular senescence can be induced prematurely by extrinsic stressors, even in younger organisms. Our focus is specifically on cellular senescence.

We do not expect the 181 CS markers to be shared across cell types, but we show that CS genes are found at lower-than-expected rates in the DGE analysis of individual cell populations

(Sup **Fig S2**) in addition to the universal organismal aging signature. To ensure clarity, we now clearly distinguish between cellular senescence and organismal aging throughout the manuscript.

- Line 123: the term "cell positivity" needs to be clarified.

Changed to "the proportion of cells expressing a gene"

- Line 136: the lack of overlap may not indicate the heterogeneity of CS markers. It may be just due to the high noise level of scRNA-seq data.

We agree that scRNA data can be noisy, especially on an individual cell level. This is one of the driving reasons why we developed *senePy* and use gene sets instead of individual genes to identify cellular senescence.

Indeed, when we consider the similarity of *senePy* gene sets, there is more (but still limited) overlap compared to overlap based on "naïve" literature-based genes. However, in the context of line 136, we are considering each cell type on a population level which lessens the effect of individual cell variability. We still believe the main driving force of the observed heterogeneity is biological, not technical. Nonetheless, we agree this is an important point and have added a note in the study limitations. Later, when we find universality between *senePy* genes, we employ a statistical model to help account for noise (discussed more below).

- Line 149: it is not clear if the proposed method for defining cell type-specific signatures is better than the method in Zhang et al. eLife 2021.

The Zhang et al. approach is complementary to the *senePy* approach because one approach identifies organismal aging (Zhang et al.) whereas *senePy* identifies cellular senescence gene signatures. We have added some additional analyses comparing both in the results section: "**Cell-specific signatures are unique but share common stress response and inflammatory pathways.**" We have added supplemental figures **S2D-F**:

20% of the DE-derived aging signatures are enriched for known senescence markers compared to 32% for our cellular senescence signatures. We also compare a universal organismal aging signature derived by differential expression (Global aging genes: GAG) to our *senePy* universal signature (**Fig S2E**). Strikingly, the *senePy* universal signature is highly enriched for known cellular senescence markers ($p = 1.4 \times 10^{-13}$) and the DE-derived signature is not. Interestingly, there is less overlap between the DE GAG set and the universal *senePy* signature than what is expected (**Fig S2F**), indicating that these methods are uncovering biologically distinct phenomena (i.e., organismal aging vs. cellular senescence).

Fig S2D**Fig S2E****Fig S2F**
- Line 152-513: is there a reference for this prior knowledge: "senescent cells accumulate with increasing organismal age but remain a minority population"

We have included a reference: [10.18632/aging.102903](https://doi.org/10.18632/aging.102903)

- Line 149: the authors should provide more details and justification for the proposed method for defining cell type-specific signatures. The procedures appear ad-hoc to me. How are the networks constructed? How are the CS signatures selected from the network of hubs? Why is this method preferred over DGE, which is more conventional? Systematic benchmarking experiments comparing the proposed method with existing methods are needed.

Differential expression is more suited for global changes in gene expression within a population and does not account for different programs within the same cell population. Since cellular senescence only afflicts a small number of cells in a given population, the senescence signature is diluted by the majority of cells. We have compared our method to DGE done in the same dataset (*Tabula muris senis*) and shown that our method is better at finding senescence markers. For example, in the DGE analysis, *Cdkn2a* was not found to be DE in any of the tested cell types. DGE is better at finding transcriptional shifts associated with organismal aging but is less suited for unenriched *in vivo* cellular senescence. Moreover, we used multiple independent *in vivo* datasets to validate this methodology (including an additional dataset in revision).

However, we improved our signature derivation methodology in this revision based on feedback from the reviewers (see schematic below). We removed the hard thresholds used in the first step of signature derivation and added statistical tests based on billions of permutations. Networks are still constructed based on co-expression. There were minor differences in the *senePy* gene signatures but the results and validation remained stable when compared to the hard thresholds. We have also included an additional validation in the revision (**Fig 6 F-L**).

- Line 154: what's a "cell-by-gene positivity matrix"? I think it refers to the binarized count matrix but it should be clarified here. Also, the authors need to provide justifications for binarization. Why is the better than the original count matrix?

The reviewer is correct in this assumption and we have updated the text accordingly. Our rationale for binarization is: 1) We wanted to give equal weight to cellular senescence markers even if they had low levels of expression because many important regulators of senescence are lowly expressed, e.g., *Cdkn2a*. 2) Senescence markers may be lowly expressed, and expression is noisier at the low end. For example, binarization gives the same weight to a cell with 1 *Cdkn2a* count as a cell with 3 counts. 3) Only a small proportion of cells will be senescent in a given population and cellular senescence is a distinct cell state, therefore reflecting a binary state. 4) Binarization accounts for differences in normalization and transformation between studies, users, etc.

- Line 161: how can we be sure that this negative result is not just because of noise? The lack of overlap seems to be a negative result, and, in general, statistical analysis cannot draw conclusions about negative results (in other words, $P < 0.05$ indicates significant effects but $P > 0.05$ doesn't indicate no effects).

We believe this is analogous to comparing marker genes between single-cell populations and inferring heterogeneity. For example, it is routine to infer that markers between T-cells and fibroblasts have little overlap. For this to be considered noise, we would have to assume that there are massive fundamental differences in how these cell types are processed and analyzed. To reduce the confounding impact of noise in our study, we only compare young and old cell populations together from the same study and platform.

- Line 162-164: this may be due to the batch effect between tissues. Have the authors verified that this is not the case?

In *tabula muris senis*, tissues were processed similarly within the same study, mitigating batch effect between tissues. We can assume gene expression between distant cell types is more dissimilar than it is for the same cell type between similarly processed batches (i.e., biological signal is stronger than a batch effect). If the observation is due to a batch effect, we would have to assume the batch effect is stronger than distant cell-type differences. It is harder to make this

claim between human studies because it is a combination of studies. We have clarified that these statistical tests were performed in *tabula muris senis* and have included a note in the limitations on batch effect.

- Line 165-166: can you quantify how prevalent this observation is? Anecdotal examples are helpful but a rigorous quantification is necessary.

Every line that connects a dot to a different column in Figure 2 represents this observation. Each signature is connected to its most similar partner. This is done for every novel signature in our study.

- Line 176: the authors concluded that "the previously defined set of unspecific senescence markers was more closely associated with the newly derived signatures". However, it is not clear what the statistical analysis is and what's the strength of evidence, such as a p-value.

This analysis is pairwise enrichment between signatures followed by Louvain clustering. All edges are statistically significant. There are more *senePy* signatures enriched for the literature marker set compared to the organismal aging signatures. However, we now mention this observation with Fig S2D and have removed this line due to redundancy.

- Line 181-183: it is not clear why *SenePy* is novel. How it is different from applying the Seurat cell-scoring module to the set of signatures identified by this paper? Also, there are alternative cell-scoring methods with better calibration, such as scDRS (Zhang et al. Nat Genet 2022).

SenePy is a novel database of cell-specific cellular senescence signatures packaged with a scoring algorithm. ***SenePy* is not just a gene-scoring method.** Additionally, the scoring method weighs gene contributions by their respective network centrality. *SenePy* also contains multiple ancillary tools to search the database, to create universal gene sets from individual signatures, harmonize gene names between studies, etc. The main emphasis of our work was to generate, validate, and apply these senescence signatures. As noted, multiple groups have already tackled the problem of scoring based on a gene set. We have incorporated an additional flavor of scoring based on scDRS into our package. By incorporating scDRS, we can likewise provide a centrality-weighted senescence score to every cell. scDRS also provides an FDR-corrected p-value for each respective cell scoring. Cells identified by scDRS have high, but not complete, overlap with cells identified by the original *senePy* method. These analyses and results are described in greater detail below (**Fig S6**).

- Line 574-578: did the authors consider confounding when computing the gene-gene correlation?

We attempted to minimize confounders at every step of the analysis. Specifically for gene-gene correlation, we only considered cells from old organisms to account for differences between old and young cells. We also only computed correlations between cells from the same study to account for differences between studies. Our updated methodology (schematic/described above) also accounts for confounders introduced by gene expression sparsity immediately prior to network construction.

- Line 192-193: Please provide p-values for the finding "For example, aging mouse tongue keratinocytes consisted of two hubs, both enriched in established CS marker genes". In general, all findings need to be supported by statistical evidence.

P-value for enrichment added to text.

- Line 192: why did you choose mouse tongue keratinocytes? How representative is this example? Anecdotal examples are helpful but a systematic quantification of the overall patterns across cell types will provide deeper insights.

The keratinocytes were chosen for two primary reasons: 1) there is more than one distinct hub in the signature and 2) there is a large number of keratinocytes across the tested ages, which helps to account for age-to-age noise. Additionally, both hubs were highly enriched for known senescence markers. Other cell types had similar temporal trends and network topology, but it is not representative of all hubs. This has been clarified in the text and we have added additional descriptive analysis of every mouse hubs into the supplement and text of the manuscript. New figures **S4A** ,**C** show the network structure of every signature and the proportion of cells expressing these signatures in young and old mice in the respective cell type. Figure **3A** shows that there is a general increase in the cells expressing these signatures with age.

Fig S4A

Fig S4C

Fig 3A

- Line 206: how was the outlier threshold determined? Would FDR be more appropriate? I note

that alternative cell-scoring methods such as scDRS (Zhang et al. Nat Genet 2022) can produce cell-level p-values.

We thank the reviewer for this comment and believe it has improved the manuscript in revision. In this instance, a standard deviation of three was determined by Chebyshev Inequality, which states that at least ~95% of data points would be lower than 3 standard deviations above the mean. We did some additional testing of these thresholds in comparison to scDRS and have incorporated these findings into the results section of the manuscript. scDRS provides a multiple comparison-corrected p value for each scored cell. In general, the cells identified by senePy at every threshold had very significant overlap with the cells identified by scDRS. Std thresholds 2 and above identified fewer cells than scDRS, but with very high overlap. This may indicate that >2 std thresholds have high specificity but potentially at the cost of sensitivity. Moreover, the binary method of senePy scoring provides bimodal distributions that make it possible to derive an empirical threshold. Nonetheless, relative comparison using a consistent threshold will yield interpretable results, regardless of the threshold chosen. When identifying absolute senescent cell counts, the user can adjust the threshold based on prior biological knowledge or empirically from the observed threshold.

Intersection between senePy outliers and scDRS significant cells may provide even more specific cell identification than either method alone. Therefore, we incorporated scDRS into the senePy toolkit so users can score cells with an alternative method that provides an FDR value.

Supplemental Figure 7

- Line 207: can you provide a p-value for "increased significantly"?

We added p-values calculated using a Chi-Square test for independence. The text has been updated, and the code can be found in notebook 7.1.

- Line 212: how was fibroblasts chosen as an example? How representative is this example? Anecdotal examples are helpful but a systematic quantification of the overall patterns across cell types will provide deeper insights.

Fibroblasts were chosen specifically because they are a cell type found in multiple tissues. The goal of this specific analysis was to explore how similar cells undergo senescence in different tissues. As mentioned above, we have added additional descriptive analysis of all 75 mouse hubs into the supplement and text of the manuscript.

- Line 230: the identification of universal senescence signatures may be confounded by the composition of cell types in your datasets.

We agree with this statement and include this as one of our limitations in our discussion: "The comprehensiveness of our signature panel is also limited by the data available at the time of study and the exact set of tissues and cells tested influences any conclusions of universality or comparison between species."

Furthermore, when determining a universal signature, we use a statistical model instead of basing it strictly on the presence of genes in a set number of signatures. This helps to account for cell-type bias and batch effect when determining the universality of a gene.

- Line 230: how do the universal senescence signatures identified here compare to the 330 global aging genes in Zhang et al. eLife 2021?

Interestingly, there is no overlap between the 635 genes of the universal *senePy* signature and the 330 global aging genes identified by Zhang et al. This is statistically underrepresented based on the background gene set (expected = 10.4, $p = 2.3 \times 10^{-5}$). Our understanding is that this is for the same reason we mentioned earlier: organismal aging and cellular senescence are distinct biological processes and our methodology is tailored to uncover cellular senescence. Conversely, and importantly, the universal *senePy* signature is enriched for known markers of cellular senescence ($p = 1.4 \times 10^{-13}$). We have updated the manuscript accordingly.

Fig S2D**Fig S2E****Fig S2F**
- Line 234: what is n in " $n \geq 12$ "? How was the threshold 25% and the p-value threshold 0.0014 determined?

" $n \geq 12$ " refers to the number of signatures a gene is found in. For example, *Cdkn2a* was found in 12 of the senePy signatures. The choice of 25% is somewhat arbitrary. Others have used such thresholds, for example, Zhang et al. eLife 2021 used a 50% signature cutoff to classify global aging genes. In this case, 25% corresponds to a multiple testing corrected p-value of 3.1×10^{-10} ; i.e., a gene is very unlikely to be found in 12 signatures by random chance. More genes are supported by a basement FDR threshold of 0.05, but 0.05 is also arbitrary, and having a stricter threshold increases universality and significance. This threshold is purely descriptive and not used in downstream analysis.

The algorithm can be viewed in notebooks 5.1 and 6.1 as well as in `senepy.load_hubs().merge_hubs(calculate_thresh = True)`. We have added a new section in the methods that covers this in greater detail (see equation 14). Of note, we have changed the random permutation method used previously to a probabilistic model that is able to calculate the p-values more accurately and to higher float precision. Importantly, we also noticed an error in the previous code that caused the permutation-based p-values to be higher, therefore leading us to underestimate the gene enrichment significance. Genes are actually more unlikely to be found in multiple hubs by random chance—further supporting the universal nature of some of the observed genes. We have also added multiple-testing correction on the resulting p-values.

- Line 241: how was the threshold of 8 chosen?

This threshold was chosen similarly to what is described above. However, the 25% threshold mentioned above was not used in downstream analysis and was only for descriptive purposes. A threshold of 8 is still highly significant (FDR p-value = 1.1×10^{-5}) but allows for more genes to be considered for gene set enrichment. Before we changed the p-value calculation this corresponded to a p-value of 0.01. We have changed the gene set enrichment to only include genes with an FDR p-value of < 0.05 to increase consistency. Many of the same pathways were enriched, but the rank and p-values differed based on the updated list; therefore, some pathways may no longer be seen on the top 15 KEGG plot (**Fig 4b**).

- Line 251: how consistent are the universal senescence signatures between human and mouse data?

There are 46 genes shared between the universal senescence signatures. This overlap is only mildly enriched (Hypergeometric $p = 0.09$). CDKN2A, CCL3, and CXCR2 are known marker genes present in both universal signatures. This was described previously in the last paragraph of the results section titled "Cell-specific signatures are unique but share common stress response and inflammatory pathways".

- Line 256: multiple testing correction is needed instead of a p-value threshold of 0.05.

We have added multiple testing correction to this and other parts of this section.

- Line 283-284: Have the authors considered confounding for explaining the between-patient difference? The lack of overlap seems to be a negative result, and, in general, statistical analysis cannot draw conclusions for negative results (in other words, $P < 0.05$ indicates significant effects but $P > 0.05$ doesn't indicate no effects).

All patients referred to here were from the same study, and the cells were processed similarly. We expect the differences observed between individuals to primarily be from true heterogeneity between individuals.

- The paper created many different types of gene signatures and it is very easy to get lost. I suggest creating a table containing all signature types, such as the 181 experimentally validated CS marker genes (line 107), aging signatures from Zhang et al. eLife 2021, and the cell type-specific signatures identified by this paper.

Our signatures and their metadata are packaged into senePy, but we agree that including them as supplemental tables will increase visibility for people who are not familiar with Python. We have now added a table containing all cell-type specific signature genes to the manuscript supplement. We have also added information on how to access all signatures in senePy using R to our GitHub with new examples. The 181 CS markers are already packaged into the senePy package, but we now provide additional examples.

Minor:

Line 107: "maker"  "marker"

Fixed

- Line 154: "matrixes"  "matrices"

Fixed

Reviewer #4 (Remarks to the Author):

Abstract

The final paragraph is too technical and should be rewritten for clarity for the broad audience of Nature Communications.

This has been corrected.

Introduction

Heterogeneity is difficult to only characterize via CDKN2A. Reporter mice for CDKN1A, for instance, are also gaining popularity in the senescence field, and the role of Cdkn2a is questioned in specific conditions and locations. Perhaps there are more markers than only these two.

This is one of the primary motivations for this work. Our literature-derived marker set has over 100 genes, and individual *senePy* signatures contain up to a few thousand genes based on the cell type. We hope that *senePy* will help users recognize the diversity of cellular senescence programs across distinct cell types, likely with varying “ideal” marker gene sets.

Results

Not finding a universal senescence marker gene set does not automatically show that there are none for specific cell types that may or may not be tissue-specific.

We agree that individual cell types that are found in multiple tissues, such as endothelial cells, fibroblasts, or macrophages, may have their own “universal” signature. We have done additional analyses of endothelial cells, fibroblasts, and macrophages within the *senePy* mouse signatures. There were 14 genes found in all the fibroblast signatures (q-value = 2.3×10^{-14}). *Nmes1* (AA467197) was found in all endothelial cell populations (q-value = 0). There were four genes found in 80% of the macrophage signatures but not all. We have added these results to the text and as a supplemental table. Additionally, *senePy* now has a function that finds genes that are statistically overrepresented in signatures defined by the user: see new methods section “Merging multiple signatures and identifying a universal senescence signature”.

What is the biological rationale for similar gene expression signatures more likely to be found in cells from the same tissue vs. cell type?

We think this is because the cell types within a tissue are exposed to similar stressors and environments, thereby predisposing them to similar senescence pathways. This effect may be subtle, yet the signatures overall are very heterogeneous, thereby making this signal easier to detect.

The increase of the *SenPy* senescence score with aging in Fig. 3E is quite convincing.

Temporal kinetics can be assessed via RNA velocity, as shown by Manno et al. (Nature 2018). This would be a perfect (and the only unbiased) way to demonstrate the superiority of your senescence signature within scRNA-Seq signatures.

We thank the reviewer for this comment and have added new analyses to the manuscript. These have been outlined in more detail below.

The demonstration of each pathway, like in Fig. 4B, does not substantially contribute to the overall story.

We believe that **Fig. 4b** and other signature enrichments are important to show that these signatures are supported by actual biological phenomena and statistical testing. For example, if the signatures were random sets of genes, they would not be strongly enriched for known

pathways. Moreover, they are enriched for sets that make intuitive biological sense for cellular senescence: inflammation, NFkB, etc.

line 292-295 and 302-303: Without experimental evidence, this statement is far beyond the verification provided.

292-295: removed.

302-303: We have adjusted the wording.

The p16+ cells association is also a huge limitation. Maybe SenePy just identifies p16-associated senescence reliably?

We agree with the reviewer. But there is a paucity of ground-truth single-cell cellular senescence datasets. We agree that additional ground-truth datasets would be great to test, but they do not exist. Since *senePy* was developed in a marker-independent approach, we believe that it would not be biased towards p16-associated senescence. We also test the *senePy* signatures in multiple *in vitro* and *in vivo* bulk RNA contexts as additional verification in p16-independent datasets. In this revision, we have included another verification dataset/analysis based on *in vivo* oncogene-induced senescence in conjunction with senescence trajectory analysis (Described in more detail below).

The spatial datasets do not add substantially to the overall story. In particular, why is COVID-19 important? The association with cellular senescence is well-known.

We believe it is important to showcase that *senePy* can be used in disease contexts to find biologically relevant results. We have removed the paragraph discussing COVID-19 from the discussion to focus the interpretation.

The spatial datasets are doubly important because they support the idea of an *in vivo* bystander effect and spatial clustering of senescent cells. The idea that senescent cells can induce cellular senescence in neighboring cells has been explored *in vitro* but not *in vivo*. While these data do not definitely indicate that there is cell-to-cell signaling, they do fulfill the requirement that the cell cluster spatially. If the senescent cells weren't clustered, then the bystander effect is not likely to be relevant within tissues.

COVID-19 datasets were chosen because of the robust literature on SARS-CoV-2 inducing premature senescence and this allowed for additional validation of our method in a translationally relevant disease setting.

The kinetics of cellular senescence has not been elucidated. Particularly, velocity should be used in the *tabula muris senis* dataset to verify that SenePy can identify a progress of cellular senescence along time (and age) in a specific cell type (/tissue).

We agree that a kinetic analysis of cellular senescence would add to the manuscript. However, *Tabula muris senis* is not well suited for this because most cells are at a steady state and not on a cellular senescence trajectory. We therefore used a recently published dataset (Chan et al. *Nature* Aug 2024) that utilized an inducible RAS model to examine *in vivo* oncogene-induced senescence in mouse hepatocytes. This dataset captures hepatocytes along their progression toward a senescent state over 30 days. Their method generated substantial numbers of senescent cells and their data are ideally suited for using *senePy* to assess cellular senescence

trajectories. In addition to our other in vivo validation, we now include analysis of this dataset to verify that *senePy* can capture the progression of cellular senescence.

Chan et al. induced cellular senescence in mouse hepatocytes and harvested them during peak senescence at 12 and 30 days, along with tumor and healthy tissue at 218 days (Fig 6F). The cells follow multiple CS pseudotime trajectories from the mV (mVenus) control root (Fig 6G). The *senePy* mouse hepatocyte signature is composed of two distinct hubs (hepatocyte 0 and hepatocyte 1). Both hubs strongly correlate to distinct pseudotime trajectories (Pearson's $p = 2 \times 10^{-232}$, $p = 0$), supporting the idea that a single cell type can have multiple CS phenotypes (Fig 6H). The universal *senePy* signature score is significantly correlated to global pseudotime ($p = 0$) (Fig 6I). Cells scored using traditional CS markers and the *senMayo* gene set were not as strongly associated with CS pseudotime (Fig 6J). The *senePy* score was inversely correlated to the hepatocyte marker, Albumin, likely due to reduced cell identity associated with cellular senescence (Fig 6K). Independent of pseudotime, *senePy* scores were higher in senescent hepatocytes relative to the control than cells scored with traditional CS markers (Fig 6L). There was a 4.3-fold increase in the mean *senePy* score at day 12 (Mann Whitney, $p = 2.9 \times 10^{-145}$) and 4.1-fold increase at day 30 ($p = 3.7 \times 10^{-177}$) relative to the control. Conversely, CS marker-based scoring did not increase at day 12 and had a 1.3-fold increase at day 30 ($p = 1.6 \times 10^{-17}$). These results indicate that *senePy* cell-specific signatures and its derived universal signature robustly recapitulate in vivo CS within mouse hepatocytes. Overall, this is another important validation of the *senePy* signatures and the scoring method.

Discussion

The discussion needs to be shortened substantially. While in the upcoming era of

senolytics, SenePy may be useful, you failed to prove that it is also worth considering in a senolytic population.

We have significantly revised the Discussion. Multiple datasets we used in validating *senePy* were obtained from mouse models of senolysis. In these data, we show that *senePy* signatures reflect transcriptional changes that occur after senolytic treatment. Moreover, many of the genes found within *senePy* signatures are known senolytic targets, e.g., the BCL2 family. We believe that 1) *senePy* can be used to validate the efficacy of senolytic treatment by analyzing the transcriptome and 2) that it likely contains other senolytic targets that have not been explored. However, we agree that this is not the focus of our work and we removed this section in the discussion.

Overall, experimental validation of SenePy (i.e. with *in vitro* cellular senescence over a long time) would help the manuscript.

We believe the new analysis of hepatocyte senescence trajectories with K-Ras dose titration provides an additional and independent experimental validation. These new data show that *senePy* may be more suitable for *in vivo* senescence analysis and that traditional methods are suitable for analyzing *in vitro* cellular senescence.

Also, a senolytic cohort that shows a reduction of SenePy would be highly appreciated.

We have tested *senePy* signatures in mouse senolytic datasets. *senePy* performs better at detecting transcriptional changes due to senolysis than other methods.

Reviewer #4 (Remarks on code availability):

The overall code looks correct.

However, the README file did not contain enough instructions for installing and running the application.

Subsequently, I was not able to install and run the code.

We have substantially improved the Github page and added additional tutorials and examples.

- Line 165-166: can you quantify how prevalent this observation is? Anecdotal examples are helpful but a rigorous quantification is necessary.
- Rv1: Still, I think a quantitative assessment is warranted. Would something like "average overlap between the same cell type from different tissues" and "average overlap between different cell types from the same tissue" work?

We agree with this recommendation. We have compared the similarity between cell types within the same tissue to cell types between tissues. We have added this to the text:

Signatures from cells within the same tissue had a higher average cosine similarity of 0.09 compared to 0.04 for those derived from different tissues (Mann Whitney, $p = 9.6 \times 10^{-7}$).

- Line 283-284: Have the authors considered confounding for explaining the between-patient difference? The lack of overlap seems to be a negative result, and, in general, statistical analysis cannot draw conclusions for negative results (in other words, $P < 0.05$ indicates significant effects but $P > 0.05$ doesn't indicate no effects).
- Rv1: Still, the pattern might be driven by patient-specific factors other than age, such as sex or dietary habits. If investigating this is challenging, it warrants further discussion.

We agree and had previously expressed it in the manuscript: "...suggesting patient-specific differences are important drivers of CS within tissues globally".

However, our ability to study the precise effects of specific confounding factors requires access to patient metadata which is not readily available. We have expanded the Discussion in the revision:

"This work may also serve as a starting point for studying how patient-specific factors such sex and lifestyle impact distinct senescent cell phenotypes and the kinetics of senescent cell accumulation."